# LoRA Recycle: Towards Fine-Tuning-Free Visual Foundation Model via Double-Efficient Data-Free Meta-Learning

## Abstract

Large Language Models (LLMs) such as ChatGPT can efficiently adapt to few-shot tasks without fine-tuning, making them ideal for data-limited applications requiring real-time responses. However, this adaptability has not yet been replicated in current Visual Foundation Models (VFMs), which require explicit fine-tuning with sufficient tuning data. Low-Rank Adaptation (LoRA), an effective fine-tuning approach, adapts VFMs to specific tasks by updating extra lightweight modules. Thanks to its modularity, users can upload locally tuned LoRAs to public repositories without exposing private training data. In this paper, we explore the potential of reusing diverse pre-tuned LoRAs without accessing their private training data, to improve the few-shot adaptability of VFMs without requiring further fine-tuning. To achieve this, we propose a data-free meta-learning framework named LoRA Recycle, which distills a meta-LoRA from diverse pre-tuned LoRAs using synthetic data generated via LoRA Inversion. The VFM, once equipped with the meta-LoRA, is empowered to solve new few-shot tasks in a single forward pass without further fine-tuning, akin to the in-context learning of LLMs. To further enhance efficiency, we propose a double-efficient mechanism that uses only the foreground patches and prunes background patches in the synthetic data, significantly accelerating the meta-training process while maintaining or even improving performance. Comprehensive experiments across eight datasets within both in- and cross-domain scenarios verify the superiority of our framework.

## 1 Introduction

Large Language Models (LLMs) like ChatGPT demonstrate a profound capacity to solve few-shot tasks without the necessity for fine-tuning, making them ideal for data-limited applications requiring real-time responses. However, this adaptability can not be replicated by current Visual Foundation Models (VFMs), which typically require explicit fine-tuning with sufficient tuning data.

To adapt VFMs to specific tasks, existing work mainly attempts to design advanced fine-tuning strategies. For instance, Low-Rank Adaptation (LoRA) (Hu et al., 2021) freezes the pre-trained model weights and injects trainable rank decomposition matrices into each layer of the Transformer architecture. While promising, (i) explicit fine-tuning is often prohibitive for applications requiring real-time responses, and (ii) fine-tuning with limited data is extremely unstable. As shown in Tab. 1, fine-tuning with limited data makes performance highly sensitive to choices like optimizer, learning rate, and step size. Besides, it also leads to significant time overheads and increased memory usage.

In the present paper, we explore the potential of reusing diverse pre-tuned LoRAs without accessing their private training data, to improve the few-shot adaptability of VFMs without requiring further fine-tuning (see Fig. 1). Our inspiration comes from the concept of *LoRA Market* (Huang et al., 2023a), where diverse pre-tuned LoRAs are publicly available and await task-specific reuse. For example, users can download and then insert a task-specific LoRA of interest into the open-source VFM, to obtain a personalized VFM. Moving beyond task-specific reuse, we seek to leverage the vast availability of these LoRAs from a novel perspective, leading to our central research question: *Is it feasible to reuse diverse pre-tuned LoRAs without accessing their private training data, to improve the few-shot adaptability of VFMs without requiring further fine-tuning?* This removes

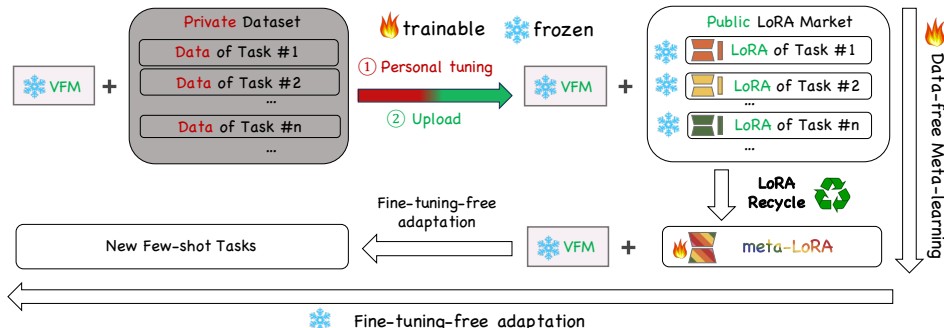

Figure 1: Thanks to the modularity of LoRA, users can upload locally tuned LoRAs to public repositories without exposing private training data. LoRA Recycle aims to distill a meta-LoRA from these LoRAs without accessing private training data. The VFM, once equipped with the meta-LoRA, is empowered to solve new few-shot tasks in a single forward pass without further fine-tuning.

the need for additional data collection, often limited by privacy concerns or high human costs, and provides insights into leveraging off-the-shelf LoRAs beyond task-specific reuse.

To answer this question, we propose a data-free meta-learning framework named **LoRA Recycle** (see Fig. 2), which distills a meta-LoRA from multiple pre-tuned LoRAs using synthetic data. Since we have no access to the original training data, we propose LoRA Inversion to generate synthetic data from the pre-tuned LoRs themselves. Then, we meta-train the meta-LoRA over a broad range of few-shot tasks constructed with these synthetic data, to explicitly learn how to adapt to diverse tasks without fine-tuning. Once equipped with the meta-LoRA, the VFM is empowered to adapt to new few-shot tasks in a single forward pass without further fine-tuning, akin to the in-context learning of LLMs. The rationale behind this is to reshape the prior of VFMs over a distribution of expected tasks, and such prior encoded in the meta-LoRA can facilitate learning of new tasks sampled from similar distributions. To further enhance efficiency, we propose a double-efficient mechanism that uses only the foreground patches and prunes background patches in the synthetic data, significantly accelerating the meta-training process while maintaining or even improving performance.

Our framework is designed to be: (i) data-free, solely using diverse pre-tuned LoRAs without requiring access to their original training data; (ii) parameter-lightweight, meta-training only 0.14M parameters in the meta-LoRA (merely 0.1% relative to the VFM); (iii) computation-efficient, achieving significant meta-training acceleration without compromising performance; (iv) architecture-agnostic, enabling to recycle LoRAs with heterogeneous architectures like different ranks, as a distinct advantage over existing methods. We outline our contributions as follows:

- **Novel perspective:** Inspired by LoRA Market, we explore the potential of reusing diverse pre-tuned LoRAs, to improve the few-shot adaptability of VFMs without re- quiring further fine-tuning. This eliminates the need for additional data collection and offers new insights into leveraging the vast availability of pre-tuned LoRAs beyond task-specific reuse.
- **Technical contributions:** (i) We propose a data-free meta-learning framework named LoRA Recycle, which distills a meta-LoRA from diverse pre-tuned LoRAs using synthetic data generated via LoRA Inversion. The VFM, once equipped with the meta-LoRA, is empowered to solve new few-shot tasks in a single forward pass without further fine-tuning. (ii) We also propose a double efficient mechanism significantly accelerating the meta-training process while maintaining or even improving performance.
- **Comprehensive evaluations:** We conduct extensive experiments across eight datasets, covering both in-domain and cross-domain scenarios. These experiments demonstrate our superiority in significantly improving the few-shot adaptability of VFMs without further fine-tuning.

## 2 RELATED WORK

### 2.1 PARAMETER EFFICIENT FINE-TUNING (PEFT)

Fine-tuning the entire foundation model results in high costs in computation and storage. To mitigate these challenges, several PEFT methods (Hu et al., 2021; He et al., 2022; Wu et al., 2023a; Liu et al., 2022; He et al., 2021; Jiang et al., 2023) have emerged, focusing on the update of a limited subset

Table 1: Fine-tuning ViT-B/16 on 600 5-way 1-shot classification tasks from the meta-testing set of CIFAR-FS. We report the accuracy, throughput (tasks per second) and GPU memory usage during fine-tuning. Values highlighted in green represent the best, whereas those in red denote the worst.

| Method | Optimizer | Step | Learning Rate | | | Throughput (tasks/s) ↑ | GPU Mem (GB) ↓ |
|--------|-----------|------|------|------|------|------|------|
| | | | 0.1 | 0.01 | 0.001 | | |
| Full Fine-Tuning | SGD | 50 | 22.81 | 30.13 | 28.99 | 0.10 | 12.88 |
| | | 5 | 20.56 | 23.69 | 23.85 | | |
| | Adam | 50 | 20.04 | 20.43 | 26.64 | | |
| | | 5 | 20.00 | 19.96 | 21.09 | | |
| LoRA | SGD | 50 | 79.29 | 77.07 | 36.61 | 0.13 | 9.54 |
| | | 5 | 73.48 | 37.27 | 20.60 | | |
| | Adam | 50 | 22.10 | 26.55 | 82.19 | | |
| | | 5 | 20.40 | 73.00 | 55.11 | | |
| **LoRA Recycle (ours)** | — | — | 89.70 (+7.51%) | | | 8.25 (×63) | 1.28 (-87%) |

of model parameters. While promising, PEFT methods underperform in data-limited and real-time scenarios due to the cost of explicit fine-tuning and the need for sufficient data. More recently, several works (Huang et al., 2023a; Wu et al., 2023a; Gou et al., 2023; Chen et al., 2024; Wu et al., 2023b) have investigated the potential of composing multiple pre-tuned LoRAs. However, (i) they mainly focus on arithmetic operations like weight averaging in the parameter space, lacking precise alignment for LoRAs targeting different label spaces in the context of classification. (ii) They are not applicable to reuse LoRAs with different architectures like different ranks.

## 2.2 META-LEARNING & DATA-FREE META-LEARNING (DFML)

Meta-learning, also known as *learning to learn*, aims to learn prior knowledge over a distribution of tasks, enabling efficient adaptation to unseen few-shot tasks sampled from similar distributions. Traditional data-based meta-learning (Finn et al., 2017; Yoon et al., 2018; Khodak et al., 2019; Fu et al., 2023a) typically assumes the availability of task-specific data for each meta-training task. Recently, DFML (Wang et al., 2021; Hu et al., 2023a;b; Wei et al., 2024b;a) emerges as a promising solution to directly meta-learn from pre-trained models available off the shelf. However, existing methods struggle to scale up to large Vision Transformers, whereas our framework only meta-trains a lightweight meta-LoRA. Additionally, we propose a double-efficient mechanism to further accelerate the meta-training process.

## 2.3 TRAINING-FREE ADAPTATION OF FOUNDATION MODELS

Compared to explicit fine-tuning, training-free adaptation requires no parameter updates, making it highly suitable for real-time applications with low computational budgets. LLMs achieve training-free adaptation through their inherent in-context learning capabilities (Dong et al., 2022). Existing studies suggest that in-context learning is equivalent to implicitly performing gradient descent (Dai et al., 2022; Von Oswald et al., 2023), viewing LLMs as meta-learning models (Brown et al., 2020). However, this in-context learning ability has not yet been replicated by current VFMs. To address this, Fifty et al. (2023) explicitly train a sequence model with VFMs to simulate LLM-style in-context learning. Zhang et al. (2024) adapt the Segment Anything Model in a training-free manner using a one-shot example. Our LoRA Recycle, on the other hand, achieves training-free adaptation to few-shot tasks from a novel perspective, by reusing diverse pre-tuned LoRAs. This avoids the need for additional data collection while leveraging the vast availability of off-the-shelf LoRAs.

## 3 PRELIMINARY & PROBLEM SETUP

**Low-Rank Adaptation (LoRA)** (Hu et al., 2021) enables VFM to solve a specific task by only updating lightweight extra modules. For a weight matrix $W^{(l)} \in \mathbb{R}^{d \times k}$ at the $l^{th}$ layer within the VFM $f$, a LoRA module is represented as a low-rank matrix decomposition $\delta W^{(l)} = \delta W_A^{(l)} \cdot \delta W_B^{(l)}$, where $\delta W_A^{(l)} \in \mathbb{R}^{d \times r}$, $\delta W_B^{(l)} \in \mathbb{R}^{r \times k}$ and the rank $r \ll \min(d, k)$. The input $\mathbf{X}_{in}$ will be processed in parallel as $\mathbf{X}_{out}^{(l)} = W^{(l)} \mathbf{X}_{in}^{(l)} + \delta W_A^{(l)} \delta W_B^{(l)} \mathbf{X}_{in}^{(l)}$. When fine-tuning, it freezes the original weight matrix $W$ while only keeping $\delta W_A$ and $\delta W_B$ trainable. When facing classification tasks, a classification head $h$ is always tuned together with the LoRA modules to output the prediction distribution. We use $f_{\delta W}$ to denote the VFM equipped with the LoRA $\delta W$.

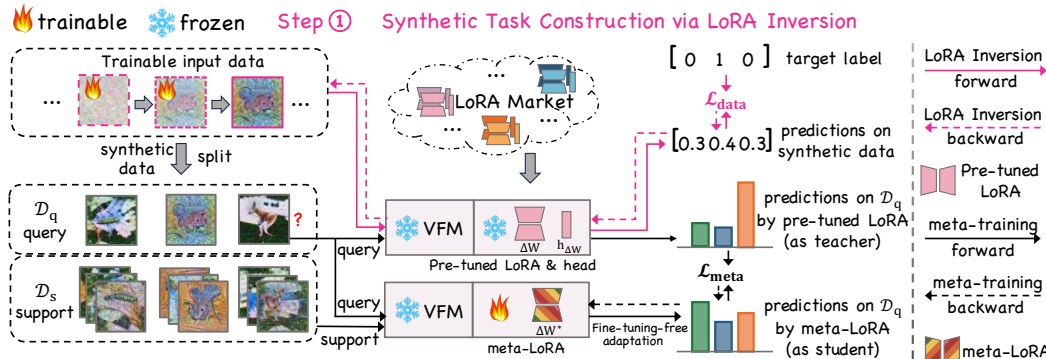

Figure 2: Pipeline of LoRA Recycle. (i) (Pink Path) We generate task-specific synthetic data from the pre-tuned LoRA via LoRA Inversion. The input data (attached with the fire in the left corner) is initialized as Gaussian noise and iteratively optimized by minimizing $\mathcal{L}_{\text{data}}$ (Eq. (1)). The synthetic data is then used to construct a few-shot task with one support set and one query set. (ii) (Black Path) We meta-train the meta-LoRA (attached with the fire in the middle) on a wide range of synthetic few-shot tasks by minimizing $\mathcal{L}_{\text{meta}}$ (Eq. (3)), explicitly teaching it how to adapt without fine-tuning.

**Problem setup.** We are given a transformer-based VFM $f$ pre-trained on large-scale datasets, and multiple LoRAs with classification heads pre-tuned on diverse classification tasks. Following standard meta-learning setup (Finn et al., 2017), we assume these tasks follow an underlying task distribution $p_{\mathcal{T}}$. $(\delta W_{\mathcal{T}}, h_{\mathcal{T}}) \sim p_{\mathcal{T}}$ denotes the LoRA and classification head pre-tuned on task $\mathcal{T}$. Note that we have no access to the original training data behind the given LoRAs. Our goal is to meta-train a meta-LoRA $\delta W^*$ over $p_{\mathcal{T}}$, so that the VFM $f$, once equipped with $\delta W^*$ (*i.e.*, $f_{\delta W^*}$), can adapt to new few-shot tasks sampled from similar distributions without further fine-tuning.

**Testing setup.** We conduct evaluation on 600 $N$-way $K$-shot classification tasks. Note that the classes in these testing tasks have not been seen by any given LoRA. Each $N$-way $K$-shot task $\mathcal{T}$ consists of one support set $\mathcal{D}_{\text{s}}^{\mathcal{T}}$ and one query set $\mathcal{D}_{\text{q}}^{\mathcal{T}}$. The support set $\mathcal{D}_{\text{s}}^{\mathcal{T}}$ has $N$ classes and $K$ examples per class. We focus on a few-shot setting where $K$ is small (e.g., 1 or 5), thus fine-tuning $f$ with extremely few examples is infeasible. In contrast, we use $\mathcal{D}_{\text{s}}^{\mathcal{T}}$ to adapt $f$ in a fine-tuning-free manner. The query set $\mathcal{D}_{\text{q}}^{\mathcal{T}}$ is what we actually make predictions on. The overall accuracy is measured by averaging the accuracy across all testing tasks.

## 4 METHODOLOGY

In this section, we present our proposed framework LoRA Recycle (see Fig. 2), which distills a meta-LoRA from diverse pre-tuned LoRAs using synthetic data. Since we have no access to original training data, we propose LoRA Inversion to generate synthetic data from pre-tuned LoRAs (see Sec. 4.1). We then meta-train the meta-LoRA over a large set of few-shot tasks constructed with these synthetic data to explicitly learn how to adapt to diverse tasks without fine-tuning (see Sec. 4.2). To further improve efficiency, we propose a double-efficient mechanism significantly accelerating the meta-training process while maintaining or even improving performance (Sec. 4.3).

### 4.1 SYNTHETIC TASK CONSTRUCTION VIA LoRA INVERSION

**LoRA Inversion.** Given a pre-tuned LoRA $\delta W$ with its classification head $h$, we synthesize its original training data by iteratively optimizing (a batch of) data $\mathbf{X}$, which is initialized as Gaussian noise. This is done by minimizing the following loss function:

$$\min_{\mathbf{X}} \mathcal{L}_{\text{data}} = \text{CE}\left(h \circ f_{\delta W}(\mathbf{X}), \mathbf{Y}\right) + \alpha_{\mathcal{R}} \mathcal{R}_{\text{BN}}(\mathbf{X}), \tag{1}$$

where $\mathbf{Y}$ is the target label (*e.g.*, $[1, 0, 0]$). $\text{CE}(\cdot)$ is a cross-entropy classification loss. $\mathcal{R}_{\text{BN}}$ is an image regularization term with a coefficient $\alpha_{\mathcal{R}}$. Minimizing the first classification loss is to achieve label-conditional generation, ensuring $\mathbf{X}$ can be predicted by $f_{\delta W}$ as the target label $\mathbf{Y}$. To further

improve the realism of the generated data, we impose a naturalness prior $\mathcal{R}_{\text{BN}}$ (Yin et al., 2020):

$$\mathcal{R}_{\text{BN}}(\mathbf{X}) = \sum_l \left\| \mu^{(l)}(\mathbf{X}) - \mu_{\text{BN}}^{(l)} \right\|_2 + \left\| \sigma^{(l)}(\mathbf{X}) - \sigma_{\text{BN}}^{(l)} \right\|_2, \quad (2)$$

where $\mu^{(l)}(\mathbf{X})$ and $\sigma^{(l)}(\mathbf{X})$ denote the mean and variance of the inputs' feature maps calculated at the $l^{\text{th}}$ layer of the pre-trained model. $\mu_{\text{BN}}^{(l)}$ and $\sigma_{\text{BN}}^{(l)}$ denote the statistics initially stored in the $l^{th}$ batch normalization (BN) layer of the pre-trained model, which is calculated with the original training data. Given that Vision Transformers do not have the BN layer, Hatamizadeh et al. (2022) suggest that we can borrow the BN statistics stored in an open-source pre-trained ResNet50. Since $\mu_{\text{BN}}^{(l)}$ and $\sigma_{\text{BN}}^{(l)}$ is calculated with real data, minimizing gaps in these statistics can align the distribution between the synthetic and real data, thus improving realism (see Fig. 9).

**Synthetic few-shot task construction.** After synthesizing a batch of task-specific data of task $\mathcal{T}$, we construct a few-shot task by splitting the synthetic data into one support set $\mathcal{D}_c^{\mathcal{T}}$ and one query set $\mathcal{D}_q^{\mathcal{T}}$. Following the standard definition of $N$-way $K$-shot, the support set has $N$ classes and $K$ examples per class, while the query set has the same $N$ classes but relatively more examples per class, typically 15. The synthetic task will be used for the following meta-training process.

### 4.2 LoRA Distillation via Meta-Learning

**Meta-learning objective.** We distill a meta-LoRA $\delta W^*$ from diverse pre-tuned LoRAs using the synthetic tasks generated via LoRA Inversion. The meta-learning objective is formulated as follows:

$$\min_{\delta W^*} \mathcal{L}_{\text{meta}} = \mathbb{E}_{(\delta W_{\mathcal{T}}, h_{\mathcal{T}}, \mathcal{D}_s^{\mathcal{T}}, \mathcal{D}_q^{\mathcal{T}}) \sim p_{\mathcal{T}}} \sum_{(\mathbf{X}_q, \mathbf{Y}_q) \in \mathcal{D}_q^{\mathcal{T}}} \text{KL} \left( P(\mathbf{Y}_{\text{pred}} | \mathbf{X}_q, \mathcal{D}_s^{\mathcal{T}}), h_{\mathcal{T}} \circ f_{\delta W_{\mathcal{T}}}(\mathbf{X}_q) \right), \quad (3a)$$

$$\text{where,} \quad P(\text{pred} = i | \mathbf{X}_q, \mathcal{D}_s^{\mathcal{T}}) = \frac{\exp\left(-\|f_{\delta W^*}(\mathbf{X}_q) - \boldsymbol{c}_i\|_2\right)}{\sum_{i'} \exp\left(-\|f_{\delta W^*}(\mathbf{X}_q) - \boldsymbol{c}_{i'}\|_2\right)}. \quad (3b)$$

Here, $(\delta W_{\mathcal{T}}, h_{\mathcal{T}}, \mathcal{D}_s^{\mathcal{T}}, \mathcal{D}_q^{\mathcal{T}})$ refer to the pre-tuned LoRA, classification head, synthetic support set, and query set of task $\mathcal{T}$, which can be viewed as sampling from the task distribution $p_{\mathcal{T}}$. The optimization in Eq. (3) involves one inner loop Eq. (3b) and one outer loop Eq. (3a).

- **Inner Loop:** We recast the inner loop Eq. (3b) as a fine-tuning-free adaptation (Snell et al., 2017): we use the support set to calculate the class center $\boldsymbol{c}_i$ of each class $i$ as the average feature embedding ($\boldsymbol{c}_i = \frac{1}{|\mathcal{D}_{s,i}^{\mathcal{T}}|} \sum_{\mathbf{X} \in \mathcal{D}_{s,i}^{\mathcal{T}}} f_{\delta W^*}(\mathbf{X})$). We then model the probability of a query example $\mathbf{X}_q \in \mathcal{D}_q^{\mathcal{T}}$ belonging to a class based on its Euclidean distance to the corresponding class center. This process does not involve any parameter updating, thus avoiding calculating any second-order derivatives (Nichol et al., 2018).
- **Outer Loop:** In the outer loop Eq. (3a), we optimize the meta-LoRA $\delta W^*$ so that it can make more accurate predictions in the inner loop. Specifically, we minimize the prediction disagreements (*i.e.*, the Kullback-Leibler (KL) divergence) on the query set $\mathcal{D}_q^{\mathcal{T}}$ between the pre-tuned LoRA $\delta W_{\mathcal{T}}$ (as the teacher) and the meta-LoRA $\delta W^*$ (as the student). The meta-LoRA $\delta W^*$ is meta-trained across a wide range of pre-tuned LoRAs sampled from $p_{\mathcal{T}}$, explicitly learning to how to solve diverse tasks without fine-tuning.

**Cross-task interpolation.** Eq. (3) assumes a diverse task distribution $p_{\mathcal{T}}$, crucial for enhancing generalization by meta-learning. However, a fixed number of LoRAs may not fully capture this diversity, especially with limited LoRA budgets. We propose cross-task interpolation that generates new tasks by combining classes from different pre-tuned LoRAs. For example, given LoRAs $\delta W_i$ and $\delta W_j$ tuned on tasks with classes (*husky*, *sparrow*) and (*golden retriever*, *wild horse*), an interpolated task might be (*husky*, *golden retriever*). This expands the range of tasks for meta-training, enhancing generalization. Since the interpolated task does not match the label spaces of any pre-tuned LoRAs, we modify Eq. (3a) by replacing the KL loss with the Cross Entropy (CE) loss:

$$\min_{\delta W^*} \mathbb{E}_{(\mathcal{D}_s^{\hat{\mathcal{T}}}, \mathcal{D}_q^{\hat{\mathcal{T}}}) \sim p_{\hat{\mathcal{T}}}} \sum_{(\mathbf{X}_q, \mathbf{Y}_q) \sim \mathcal{D}_q^{\hat{\mathcal{T}}}} \text{CE} \left( P(\mathbf{Y}_{\text{pred}} | \mathbf{X}_q, \mathcal{D}_s^{\hat{\mathcal{T}}}; \delta W^*), \mathbf{Y}_q \right), \quad (4)$$

where $p_{\hat{\mathcal{T}}}$ refers to the interpolated task distribution and $(\mathcal{D}_s^{\hat{\mathcal{T}}}, \mathcal{D}_q^{\hat{\mathcal{T}}})$ refer to the synthetic support and query sets of the interpolated task $\hat{\mathcal{T}}$.

Figure 3: Double-Efficient Mechanism. During LoRA Inversion (upper part), we prune unimportant tokens in the hidden layers. During meta-training (lower part), we transform the generated image into a masked version by multiplying the image with a mask matrix. We only feed forward the unmasked tokens for meta-training, significantly reducing computational complexity of meta-training.

## 4.3 DOUBLE-EFFICIENT MECHANISM

**Efficient data synthesis with token pruning.** As shown in Eq. (1), synthetic data generation via LoRA Inversion requires iteratively optimizing all pixels in the input data $\mathbf{X}$. To improve efficiency, we propose to prune unimportant tokens during the inversion process. This is reasonable since foreground tokens typically are more informative than background tokens. As illustrated in the left panel of Fig. 3, at the $i^{\text{th}}$ layer, we implement "token pruning" by directly discarding those unimportant tokens, no longer processing them forward or computing backward gradients, thus significantly reducing computational complexity.

*The most important tokens are those with highest attention weights in $\boldsymbol{a}_{\text{[CLS]}}$.* Suppose we have $n+1$ tokens $[\boldsymbol{x}_{\text{[CLS]}}, \boldsymbol{x}_1, ..., \boldsymbol{x}_n]$ at the $i^{\text{th}}$ layer, where $\boldsymbol{x}_{\text{[CLS]}}$ is the class token inserted before all image tokens to grasp global information. We propose to use the attention weights of the class token $\boldsymbol{x}_{\text{[CLS]}}$ with respect to all other tokens, as an indicator measuring each token's importance:

$$\boldsymbol{a}_{\text{[CLS]}} = \text{Softmax}\left(\frac{\boldsymbol{q}_{\text{[CLS]}} \cdot \boldsymbol{K}^{\top}}{\sqrt{d}}\right), \tag{5}$$

where $\boldsymbol{a}_{\text{[CLS]}}$ is a $(n+1)$-dimension vector, representing the attention weights from token $\boldsymbol{x}_{\text{[CLS]}}$ to all tokens $[\boldsymbol{x}_{\text{[CLS]}}, \boldsymbol{x}_1, ..., \boldsymbol{x}_n]$. $\boldsymbol{q}_{\text{[CLS]}}$ is the *query vector* of token $\boldsymbol{x}_{\text{[CLS]}}$. $\boldsymbol{K} = [\boldsymbol{k}_{\text{[CLS]}}, \boldsymbol{k}_1, ..., \boldsymbol{k}_n]^{\top}$ is the *key vectors* of all tokens. $d$ is the dimension of the *query vector*. The $\boldsymbol{a}_{\text{[CLS]}}$ is then used to calculate the output of token $\boldsymbol{x}_{\text{[CLS]}}$ via the self-attention mechanism:

$$\boldsymbol{x}_{\text{[CLS]}} = \boldsymbol{a}_{\text{[CLS]}} \cdot \boldsymbol{V}, \tag{6}$$

where $\boldsymbol{V} = [\boldsymbol{v}_{\text{[CLS]}}, \boldsymbol{v}_1, ..., \boldsymbol{v}_n]^{\top}$ is the *value vectors* of all tokens. Therefore, the output of $\boldsymbol{x}_{\text{[CLS]}}$ can be viewed as a linear combination of all tokens' *value vectors* weighted by $\boldsymbol{a}_{\text{[CLS]}}$. Since the output of $\boldsymbol{x}_{\text{[CLS]}}$ is used for classification at the final layer, it is rational to view $\boldsymbol{a}_{\text{[CLS]}}$ as an indicator, measuring the extent to which each token contributes to final predictions, *i.e.*, the importance of each token. Therefore, we identify the most important tokens as those with the highest attention weights in $\boldsymbol{a}_{\text{[CLS]}}$. For multi-head self-attention, we compute average attention weights $\boldsymbol{a}_{\text{[CLS]}}$ across all heads. Note that this process requires no extra computational demands, as it is an inherent part of the feed-forward process in transformers (see App. B for more preliminaries).

**Efficient meta-training with sparse tokens.** After token pruning, we obtain the most important tokens at the last layer. Since each token (except for token $\boldsymbol{x}_{\text{[CLS]}}$) precisely corresponds to a patch area in the input image, these remaining tokens can indicate which areas are more important.

*Mask construction.* As shown in the right panel of Fig. 3, we construct a mask matrix to highlight the most important areas in the generated image and mask unimportant areas. The mask matrix is constructed by setting values of 1 at the positions of remaining tokens and 0 elsewhere.

*Meta-training with sparse tokens.* We multiply the mask with the generated image to create a masked image, highlighting the important areas (such as foregrounds). When meta-training the meta-LoRA, we only feed forward the unmasked tokens, significantly speeding up the meta-training process. Our visualization results (Fig. 4) demonstrate that the unmasked areas typically correspond to the foregrounds while masked correspond to the backgrounds. Interestingly, the findings in Tab. 2 and Tab. 3 suggest that masking the backgrounds can lead to performance improvements.

### 4.4 OVERALL ALGORITHM

**Meta-training stage: LoRA Recycle.** We summarize our proposed LoRA Recycle in Alg. 1.

---

**Algorithm 1:** LoRA Recycle

---

1 **INPUT** Then VFM $f$. Multiple pre-tuned LoRAs and classification heads. Coefficient $\alpha_{\mathcal{R}}$ in Eq. (1).
2 **OUTPUT** The meta-trained meta-LoRA $\delta W^*$
3 Randomly initialize the meta-LoRA $\delta W^*$
4 **while** *not done* **do**
5      **if not** cross-task interpolation **then**
6          Randomly sample a LoRA and head $(\delta W, h_{\delta W})$
         `// Synthetic task construction via LoRA Inversion`
7          Equip $f$ with sampled LoRA and head $(\delta W, h_{\delta W})$
8          Generate synthetic data via LoRA Inversion by minimizing Eq. (1)
9          Transform data into masked versions
10          Construct a few-shot task by splitting data to one support set and one query set $(\mathcal{D}_s^{\mathcal{T}}, \mathcal{D}_q^{\mathcal{T}})$
         `// LoRA distillation via meta-learning`
11          Equip $f$ with the meta-LoRA $\delta W^*$
12          Make predictions on query examples $\mathcal{D}_q^{\mathcal{T}}$ based on support examples $\mathcal{D}_s^{\mathcal{T}}$ via Eq. (3b)
13          Update $\delta W^*$ by minimizing Eq. (3a)
14      **else**
         `// Cross-task interpolation`
15          Construct the interpolated task $(\mathcal{D}_s^{\hat{\mathcal{T}}}, \mathcal{D}_q^{\hat{\mathcal{T}}})$
16          Equip $f$ with the meta-LoRA $\delta W^*$
17          Make predictions on query examples $\mathcal{D}_q^{\hat{\mathcal{T}}}$ based on support examples $\mathcal{D}_s^{\hat{\mathcal{T}}}$ via Eq. (3b)
18          Update $\delta W^*$ by minimizing Eq. (4)

---

**Meta-testing stage: Fine-tuning-free adaptation.** After meta-training, we obtain meta-trained meta-LoRA $\delta W^*$. The testing task $\mathcal{T} = (\mathcal{D}_s, \mathcal{D}_q)$ consists of one support set and one query set. The support set is used as "context examples" to adapt the VFM $f$ to the specific task, while the query set is what we actually predict. To predict the label of each query example $\mathbf{X}_q \in \mathcal{D}_q$, we first equip the VFM $f$ with the meta-trained $\delta W^*$ to obtain the enhanced VFM $f_{\delta W^*}$. Then we feed forward the support set $\mathcal{D}_s$ and the query example $\mathbf{X}_q$ into $f_{\delta W^*}$. We directly output $p(\mathbf{Y}_{\text{pred}} = i|\mathbf{X}_q, \mathcal{D}_s)$, the probability of $\mathbf{X}_q$ being classified to label $i$ via Eq. (3b) without any fine-tuning. We assign the label with the max probability as the prediction result.

## 5 EXPERIMENTS

In this section, we perform comprehensive experiments on eight datasets, covering both in-domain (see Sec. 5.1) and cross-domain scenarios (see Sec. 5.2). We also provide comprehensive visualization results and ablation studies in Sec. 5.3 and App. A.

**Setup of VFM.** We select the 12-layer ViT-B/16 and ViT-B/32 pre-trained with CLIP as the pre-trained VFM, publicly available on HuggingFace.

**Baselines.** We compare LoRA Recycle against several baselines (see App. D for more details).

- **Fine-tuning baselines.** "Full Fine-Tuning" updates the entire model on the target task via gradient descent. "Linear probe" only updates the classification head. "LoRA + Linear (Hu et al., 2021)" updates the layer-wise rank decomposition matrices and the classification head.
- **Multi-LoRAs composition baselines.** "LoRAs Avg" refers to averaging all given pre-tuned LoRAs into a single LoRA, which can be further fine-tuned with the classification head ("LoRAs Avg + Linear") or directly make inference via Nearest Neighbour ("LoRAs Avg + NN") without fine-tuning. "LoRAHub (Huang et al., 2023a)" takes a further step, which obtains a single LoRA by a weighted sum of given pre-tuned LoRAs, where the weight values are fine-tuned on the target task. "MOLE (Chen et al., 2024)" fine-tunes a learnable gating function to composing the outpurs of different LoRAs.
- **Few-shot adaptation.** Current state-of-the-art baseline "P > M > F (Hu et al., 2022)" stacks three stages pre-training, meta-training and fine-tuning to perform few-shot adaptation.

- **Fine-tuning-free baselines.** "Nearest Neighbour (NN)" makes predictions based on the label of the closest class center. "CAML (Fifty et al., 2023)" trains a sequence model to simulate the in-context learning of LLMs.

## 5.1 RECYCLE IN-DOMAIN LoRAs

**In-domain benchmarks.** For the scenario of "recycle in-domain LoRAs", we leverage four datasets widely used in recent meta-learning works, including CIFAR-FS (Bertinetto et al., 2018), MiniImageNet (Vinyals et al., 2016), VGG-Flower (Nilsback & Zisserman, 2008) and CUB-200-2011 (CUB) (Wah et al., 2011). These datasets span a wide range, from general natural images (in CIFAR-FS and MiniImageNet) to more specialized ones, focusing specifically on different species of birds (in CUB) and flowers (in VGG-Flower). Following standard splits in the context of meta-learning (Finn et al., 2017), we split each dataset into separate meta-training and meta-testing subsets, which have non-overlapping label spaces. To construct a $N$-way $K$-shot meta-training or testing task, we randomly sample $N$ classes and $K$ examples per class from meta-training or testing subsets.

**In-domain setup.** We collect 100 LoRAs pre-tuned on diverse 5-way tasks constructed from one specific meta-training subset. Our evaluation, in contrast, is based on meta-testing tasks constructed from the corresponding meta-testing subset. This setup ensures the pre-tuned LoRAs and meta-testing tasks originate from the same domain (dataset) but with non-overlapping label spaces.

**Implementation details.** In the pre-tuning phase, we fine-tune the LoRAs with rank $r = 4$ and classification heads using the Adam optimizer with a learning rate of $1 \times 10^{-3}$. We find LoRA works quite well when fine-tuned with sufficient data. During meta-training, the meta-LoRA is optimized with the Adam optimizer, with a learning rate of $1 \times 10^{-3}$. We use a learning rate schedule that cycles every 100 iterations. It starts with a linear warm-up stage by linearly increasing the learning rate from $1 \times 10^{-5}$ to $1 \times 10^{-3}$ over the first 25 iterations. This initial increase is then followed by a cosine annealing stage, where the learning rate smoothly decreases following a cosine curve for the next 75 iterations. For LoRA Inversion, we optimize the synthetic data using the Adam optimizer with a learning rate of 0.25 for 2000 iterations. The resolution of the synthetic image is $224 \times 224$. We empirically set the hyperparameter $\alpha_{\mathcal{R}} = 0.01$. For data processing, we adopt widely used data augmentation techniques including random horizontal flip and normalization in meta-training and only normalization in meta-testing. Discussions on hyperparameter selection and sensitivity analysis are provided in App. C.

**Results and analysis.** Tab. 2 shows the results for the "recycle in-domain LoRAs" scenario. Notable findings are as follows: (i) LoRA Recycle surpasses the best fine-tuning-based baselines by considerable margins, especially up to 9.80% for 1-shot learning. It also outperforms the top fine-tuning-free baselines by up to 10.01% for 1-shot learning, confirming its superior adaptability without the need for fine-tuning. (ii) "Full fine-tuning" performs the worst, as it tends to overfit when tuning large models with extremely few examples. This issue also leads to poor performance of "P > M > F", though the second meta-training stage helps mitigate it to some extent. "LoRAs Avg" and "LoRAHub" do not ensure effective generalization to new tasks. The reason is that each pre-tuned LoRA targets different tasks, and the arithmetic operation like averaging in the parameter space lacks precise alignment among different LoRAs. Moreover, these baselines do not explicitly incorporate meta-learning objectives, which have proven to be useful for enhancing generalization in few-shot learning. (iii) Meta-training with sparse tokens can bring performance gains up to 1.34%. A reasonable explanation is that masking the backgrounds helps prevent overfitting to noise and avoids spurious correlations between the foreground and background (Ye et al., 2024). More discussions on the different performance gains across datasets are provided in App. E.

## 5.2 RECYCLE CROSS-DOMAIN LoRAs

**Cross-domain benchmarks.** Real-world situations might pose challenges in collecting LoRAs from the identical domain. For "recycle cross-domain LoRAs", we construct meta-training tasks from the meta-training subsets of CIFAR-FS, MiniImageNet, VGG-Flower and CUB. However, we construct meta-testing tasks from one specific dataset (ChestX (Wang et al., 2017), ISIC (Codella et al., 2019; Tschandl et al., 2018), EuroSAT (Helber et al., 2019) or CropDiseases (Mohanty et al., 2016)), following the standard cross-domain meta-learning benchmark (Guo et al., 2020). These datasets cover a broad spectrum, from medical images (X-rays in ChestX and skin lesion images

Table 2: Recycle in-domain LoRAs. The VFM utilizes ViT-B/16 pre-trained by CLIP. **FT** refers to fine-tuning-based baselines and **FTF** refers to fine-tuning-free baselines. **LoRA Recycle**$_x$ indicates using $x\%$ token-masked images for meta-training. The superscripts represent performance gains over the best FT baselines, while the subscripts indicate gains over the best FTF baselines.

| Method | | CIFAR-FS | | MiniImageNet | | VGG-Flower | | CUB | |
|---|---|---|---|---|---|---|---|---|---|
| | | 5-way 1-shot | 5-way 5-shot | 5-way 1-shot | 5-way 5-shot | 5-way 1-shot | 5-way 5-shot | 5-way 1-shot | 5-way 5-shot |
| **FT** | Full Finetuning | 22.81 | 28.33 | 21.16 | 23.60 | 23.11 | 31.25 | 21.27 | 24.47 |
| | Linear-probe | 80.06 | 95.49 | 82.04 | 94.12 | 89.65 | 97.77 | 85.84 | 97.40 |
| | LoRA + Linear | 79.29 | 95.43 | 82.00 | 94.83 | 88.47 | 97.63 | 85.87 | 97.32 |
| | P > M > F | 79.54 | 95.62 | 82.77 | 95.12 | 89.32 | 97.65 | 86.12 | 97.38 |
| | LoRAs Avg + Linear | 80.25 | 96.07 | 83.59 | 95.43 | 90.05 | 97.73 | 87.13 | 97.49 |
| | MOLE | 80.31 | 96.11 | 83.53 | 95.41 | 90.14 | 97.68 | 87.07 | 97.21 |
| | LoRAHub | 81.02 | 96.24 | 83.68 | 95.72 | 90.89 | 97.75 | 87.22 | 97.51 |
| | NN | 78.06 | 94.09 | 81.08 | 93.85 | 89.75 | 97.78 | 85.11 | 96.09 |
| | LoRAs Avg + NN | 79.37 | 93.45 | 81.72 | 94.64 | 90.08 | 97.92 | 85.16 | 97.23 |
| | CMAL | 81.02 | 93.59 | 81.89 | 94.81 | 91.10 | 97.98 | 86.51 | 97.32 |
| **FTF** | **LoRA Recycle** | 89.69 | $97.05^{(+0.81\%)}_{(+2.96\%)}$ | $88.60^{(+4.92\%)}_{(+6.71\%)}$ | 96.12 | $94.53^{(+3.64\%)}_{(+3.43\%)}$ | 98.59 | 91.12 | 97.67 |
| | **LoRA Recycle$_{25}$** | $91.03^{(+9.80\%)}_{(+10.01\%)}$ | 96.53 | 87.51 | $96.25^{(+0.53\%)}_{(+1.41\%)}$ | 94.38 | 98.53 | 90.16 | 97.48 |
| | **LoRA Recycle$_{50}$** | 90.91 | 96.08 | 87.21 | 95.85 | 94.05 | 98.56 | 90.65 | 97.41 |
| | **LoRA Recycle$_{75}$** | 89.70 | 96.69 | 87.36 | 96.05 | 94.28 | $98.76^{(+0.99\%)}_{(+0.78\%)}$ | $91.21^{(+3.99\%)}_{(+4.70\%)}$ | $98.23^{(+0.72\%)}_{(+0.91\%)}$ |

Table 3: Recycle cross-domain LoRAs. The VFM is ViT-B/16 pre-trained by CLIP.

| Method | | ChestX | | ISIC | | EuroSAT | | CropDiseases | |
|---|---|---|---|---|---|---|---|---|---|
| | | 5-way 1-shot | 5-way 5-shot | 5-way 1-shot | 5-way 5-shot | 5-way 1-shot | 5-way 5-shot | 5-way 1-shot | 5-way 5-shot |
| **FT** | Full Finetuning | 20.12 | 20.00 | 21.33 | 26.21 | 25.55 | 36.11 | 22.45 | 28.48 |
| | Linear-probe | 21.20 | 24.00 | 31.17 | 43.60 | 62.64 | 83.91 | 77.48 | 92.57 |
| | LoRA + Linear | 21.05 | 22.37 | 30.72 | 45.16 | 68.13 | 88.68 | 77.33 | 94.19 |
| | P > M > F | 21.12 | 22.21 | 30.77 | 45.54 | 68.51 | 88.71 | 77.65 | 94.21 |
| | LoRAs Avg + Linear | 21.37 | 20.84 | 30.51 | 45.88 | 68.77 | 88.29 | 78.75 | 94.37 |
| | MOLE | 21.24 | 20.67 | 30.61 | 45.79 | 68.84 | 88.42 | 78.81 | 94.40 |
| | LoRAHub | 21.45 | 22.61 | 32.11 | 46.12 | 69.45 | 89.76 | 79.32 | 94.44 |
| | NN | 21.23 | 22.84 | 31.20 | 40.58 | 61.73 | 80.05 | 75.48 | 91.89 |
| | LoRAs Avg + NN | 20.80 | 23.04 | 29.67 | 39.56 | 62.52 | 78.87 | 78.91 | 91.57 |
| | CMAL | 21.26 | 23.24 | 29.97 | 41.27 | 67.69 | 83.87 | 79.71 | 93.38 |
| **FTF** | **LoRA Recycle** | 22.32 | 24.61 | 33.76 | 47.96 | 66.95 | 85.17 | 83.07 | 95.33 |
| | **LoRA Recycle$_{25}$** | 22.77 | 24.88 | 33.64 | $48.29^{(+2.41\%)}_{(+7.02\%)}$ | 67.65 | 84.73 | 82.41 | 95.40 |
| | **LoRA Recycle$_{50}$** | $23.08^{(+1.63\%)}_{(+1.82\%)}$ | $25.43^{(+1.43\%)}_{(+2.19\%)}$ | $35.31^{(+3.20\%)}_{(+4.11\%)}$ | 47.41 | $69.88^{(+0.43\%)}_{(+2.19\%)}$ | 86.72 | $83.63^{(+4.31\%)}_{(+3.92\%)}$ | $96.33^{(+1.89\%)}_{(+2.95\%)}$ |
| | **LoRA Recycle$_{75}$** | 22.99 | 24.91 | 35.16 | 48.25 | 68.00 | $87.98_{(+4.11\%)}$ | 81.92 | 95.64 |

in ISIC) to specialized images (satellite imagery in EuroSAT and plant disease photos in CropDiseases). Importantly, the task domains in meta-testing are markedly dissimilar from meta-training.

**Cross-domain setup.** We collect 100 LoRAs pre-tuned on diverse 5-way tasks constructed from four meta-training subsets, including CIFAR-FS, MiniImageNet, VGG-Flower and CUB. Our evaluation, in contrast, is based on meta-testing tasks from one specific cross-domain dataset (ChestX, ISIC, EuroSAT or CropDiseases). This ensures the pre-tuned LoRAs and meta-testing tasks originate from distinctly different domains (datasets) and also with strictly non-overlapping label spaces.

**Results and analysis.** Tab. 3 shows the results for the "recycle cross-domain LoRAs". Even without fine-tuning, LoRA Recycle still outperforms the best fine-tuning-based baselines up to 4.31% and 2.41% for 1-shot and 5-shot learning, respectively. It also exceeds the top fine-tuning-free baselines by up to 4.11% and 7.02% for 1-shot and 5-shot learning, confirming its superior cross-domain robustness. The superior cross-domain performance of the meta-LoRA is attributed to its design, (i) acquiring knowledge from multiple LoRAs pre-tuned across diverse domains and (ii) benefiting from the comprehensive knowledge learned from the large-scale pre-training of the frozen VFM. These features make our approach highly ideal for deployment in real-world applications, providing significant advantages when operating across different domains.

## 5.3 ABLATION STUDIES

**How to choose the overall pruning ratio and pruning layers?** To enhance efficiency, we can alternatively adopt our proposed double-efficient mechanism. (i) The first step is to determine the overall pruning ratio (*i.e.*, sparse ratio of the synthetic data). As shown in Tab. 2 and Tab. 3, pruning overall 50% or 75% of tokens enhances the performance and tightly preserves the foregrounds (see Fig. 4). (ii) The second step is to select the pruning layers. To achieve a 75% overall pruning ratio, we could prune 75% of tokens at shallow layers or deep layers. Choosing different pruning layers only affects the speed of inversion, while the speed of the following meta-training depends solely on the overall pruning ratio. The choice of pruning layers is flexible and depends on needs: Tab. 4 suggests pruning at the deeper layer for better performance or at shallower layers for faster inversion. The choice of middle-layer pruning or multi-layer pruning across shallow and deep layers can balance the trade-off between efficiency and performance to some extent.

Figure 4: Visualization of generated images with their 75% token-masked versions.

Table 4: Complexity analysis of LoRA Inversion. $\{\mathbf{x}: \mathbf{y}\}$ denotes we prune $(\mathbf{y} \times 100)\%$ tokens at the $\mathbf{x}^{\text{th}}$ layer. Measurements are recorded during LoRA Inversion with a batch size of 25 on CIFAR-FS.

| Token Pruning Strategy | 5w 1s | 5w 5s | Throughput (its/s) ↑ | FLOPs (G) ↓ | GPU Mem (GB) ↓ |
|---|---|---|---|---|---|
| $\{\mathbf{0}: 0.0\}$ | 89.69 | 97.05 | 5.56 | 50.59 | 8.74 |
| $\{\mathbf{11}: 0.75\}$ | 89.43 | 96.72 | 5.81 (+4%) | 48.51 (-4%) | 8.63 (-1%) |
| $\{\mathbf{8}: 0.75\}$ | 82.27 | 95.69 | 6.22 (+12%) | 39.14 (-23%) | 8.07 (-8%) |
| $\{\mathbf{6}: 0.75\}$ | 81.08 | 95.52 | 7.15 (+29%) | 32.89 (-35%) | 7.69 (-12%) |
| $\{\mathbf{3}: 0.3, \mathbf{6}: 0.3, \mathbf{8}: 0.3, \mathbf{11}: 0.3\}$ | 84.17 | 96.12 | 6.13 (+10%) | 40.00 (-21%) | 8.08 (-8%) |

**Effect of sparse ratio of the synthetic data for the performance and complexity of meta-training.** As shown in Tab. 5, discarding 75% tokens can achieve up to $3\times$ acceleration for meta-training and also bring up to +0.56% performance gains on CUB. The performance gains on CIFAR-FS are up to +1.34% (see Tab. 2). A reasonable explanation is that masking backgrounds prevent overfitting to noise and avoids potential spurious correlations between foregrounds and backgrounds.

Table 5: Effect of sparse ratio of the synthetic data for the performance and complexity of meta-training. Measurements are recorded during meta-training with a batch size of 100 on CUB.

| Sparse ratio | 5w 1s | 5w 5s | Throughput (its/s) ↑ | FLOPs (G) ↓ | GPU Mem (GB) ↓ |
|---|---|---|---|---|---|
| LoRA Recycle | 91.12 | 97.67 | 1.76 | 50.59 | 12.86 |
| LoRA Recycle$_{25}$ | 90.16 | 97.48 | 2.34 (+33%) | 38.09 (-25%) | 9.40 (-27%) |
| LoRA Recycle$_{50}$ | 90.65 | 97.41 | 3.63 (+106%) | 25.60 (-49%) | 6.23 (-52%) |
| LoRA Recycle$_{75}$ | 91.21 | 98.23 | 6.83 (+287%) | 13.10 (-74%) | 3.31 (-74%) |

**Visualization.** As shown in Fig. 4, our inversion method can effectively reserve the semantic foregrounds while discarding the uninformative and noisy backgrounds. These complex generated images are of high resolution $224 \times 224$, significantly surpassing the quality of those generated by existing inversion methods (see comparisons in Fig. 7 of App. A).

## 6 CONCLUSION

In this paper, we reveal the limitations of current VFMs, which necessitate (i) explicit fine-tuning and (ii) sufficient data when adapting to new tasks. These limitations restrict their applicability in data-limited scenarios requiring real-time responses. From a novel perspective, we explore the potential of reusing diverse pre-tuned LoRAs without accessing their private training data, to improve the few-shot adaptability of VFMs without requiring further fine-tuning. To achieve this, we propose a data-free meta-learning framework named LoRA Recycle, which distills a meta-LoRA from diverse pre-tuned LoRAs using synthetic data generated via LoRA Inversion. The VFM, once equipped with the meta-LoRA, is empowered to solve new few-shot tasks in a single forward pass without further fine-tuning. To further improve efficiency, we propose a double efficient mechanism achieving significant meta-training acceleration while maintaining or even improving performance. Comprehensive experiments across eight datasets within both in- and cross-domain scenarios verify the superiority of our framework in significantly improving the few-shot adaptability of VFMs without further fine-tuning.

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

# Appendix

## A ADDITIONAL EXPERIMENTS

**Results of ViT-B/32.** Tab. 6 show the results when using ViT-B/32 with a $32 \times 32$ input patch size as the implementation of VFM. In the "recycle in-domain LoRAs" scenario, our LoRA Recycle consistently outperforms the best fine-tuning-based baselines by a large margin, up to 8.93% and 1.40% for 1-shot and 5-shot learning, respectively. It also exceeds the leading fine-tuning-free baselines by up to 10.39% and 2.89% for 1-shot and 5-shot learning, respectively. Fig. 5 shows the visualization of synthetic images and their masked versions synthesized from ViT-B/32.

Table 6: Recycle in-domain LoRAs. VFM is implemented with ViT-B/32. **FT** refers to fine-tuning-based baselines and **FTF** refers to fine-tuning-free baselines. **LoRA Recycle**$_x$ indicates $x$% tokens in synthetic data are masked (*i.e.*, different sparsity ratios). For a fair comparison between different sparsity ratios, we perform token pruning at the same layer (*i.e.*, at the last layer). Superscripts represent performance gains over the best FT baselines, while subscripts indicate gains over the best FTF baselines.

| Method | | CIFAR-FS | | MiniImageNet | | Flower-VGG | | CUB | |
|---|---|---|---|---|---|---|---|---|---|
| | | 5-way 1-shot | 5-way 5-shot | 5-way 1-shot | 5-way 5-shot | 5-way 1-shot | 5-way 5-shot | 5-way 1-shot | 5-way 5-shot |
| FT | Full Finetuning | 20.02 | 20.32 | 20.07 | 20.01 | 20.00 | 20.08 | 20.01 | 20.03 |
| | Linear-probe | 76.92 | 92.93 | 81.28 | 92.95 | 85.12 | 96.71 | 78.76 | 94.88 |
| | Lora + Linear | 76.44 | 94.85 | 79.20 | 93.60 | 83.17 | 96.57 | 76.39 | 95.43 |
| | P > M > F | 77.45 | 94.92 | 79.31 | 93.02 | 84.53 | 96.46 | 77.42 | 96.41 |
| | Loras Avg + Linear | 78.35 | 95.03 | 79.97 | 93.61 | 85.00 | 96.64 | 78.96 | 95.31 |
| | MOLE | 78.62 | 95.23 | 79.41 | 93.43 | 85.12 | 96.43 | 79.02 | 95.38 |
| | LoraHub | 79.48 | 95.36 | 80.12 | 93.93 | 85.63 | 96.69 | 79.54 | 95.48 |
| FTF | NN | 75.69 | 91.91 | 78.38 | 92.55 | 86.47 | 96.62 | 77.71 | 93.99 |
| | Loras Avg + NN | 77.05 | 92.56 | 79.63 | 92.60 | 84.21 | 96.35 | 76.32 | 93.61 |
| | CAML | 78.02 | 93.23 | 80.83 | 93.14 | 85.35 | 96.54 | 78.02 | 94.12 |
| | LoRA Recycle | 87.37 | 95.93 | 84.65 | 95.03 | $91.92^{(+6.29\%)}_{(+5.45\%)}$ | 97.65 | $85.81^{(+6.27\%)}_{(+7.79\%)}$ | $95.95^{(+0.47\%)}_{(+1.83\%)}$ |
| | LoRA Recycle$_{25}$ | 87.91 | 96.09 | 84.93 | 95.06 | 90.49 | $97.73^{(+1.02\%)}_{(+1.11\%)}$ | 84.61 | 95.73 |
| | LoRA Recycle$_{50}$ | $88.41^{(+8.93\%)}_{(+10.39\%)}$ | $96.12^{(+0.76\%)}_{(+2.89\%)}$ | $85.61^{(+4.33\%)}_{(+4.78\%)}$ | $95.33^{(+1.40\%)}_{(+2.19\%)}$ | 90.29 | 97.52 | 84.85 | 95.57 |
| | LoRA Recycle$_{75}$ | 85.99 | 95.41 | 83.75 | 94.56 | 89.89 | 97.72 | 84.09 | 95.27 |

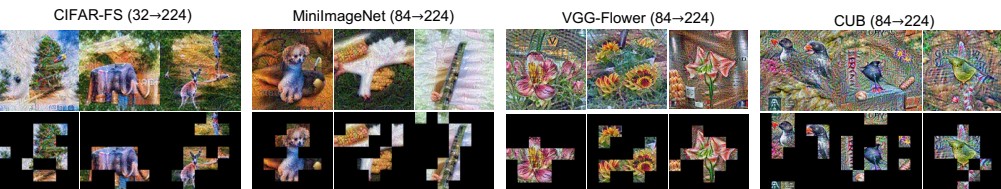

CIFAR-FS (32→224)  MiniImageNet (84→224)  VGG-Flower (84→224)  CUB (84→224)

Figure 5: Visualization of synthetic images (odd line) and their 75% token-masked versions (even line) from ViT-B/32. (32 → 224) denotes the original training images' resolution is $32 \times 32$ while we can reconstruct images with a higher resolution of $224 \times 224$. Note that the size of each patch is $32 \times 32$, instead of $16 \times 16$.

**Visualization of masked synthetic images at varying sparsity levels.** Fig. 6 illustrates synthetic images masked at varying sparsity levels. As we can see, only a subset of tokens carry meaningful semantic information and contribute to the final predictions, while the rest often represent noise, constructed as hallucinations of the VFM's misinterpretations. Our method can effectively filter out those noisy tokens and preserve the meaningful tokens, thus effectively preventing VFM from overfitting to irrelevant noise.

**Comparison with SOTA model inversion approach.** Fig. 7 illustrates that the quality of our model inversion approach surpasses current state-of-the-art (SOTA) methods like CMI (Fang et al., 2021), which typically produce simpler, lower-resolution images from shallow pre-trained models. Our approach excels in three key areas: (i) quality, producing higher fidelity images; (ii) resolution, capable of generating complex images with higher resolutions of $224 \times 224$; and (iii) efficiency, with our double-efficient mechanism significantly accelerating the model inversion process. Moreover, our work investigates the inversion from transformer-based models, whereas existing methods mainly concentrate on convolutional architectures such as ResNet.

**T-SNE visualization.** Fig. 8 presents the t-SNE visualizations of images synthesized from LoRAs pre-tuned on diverse datasets, including CIFAR-FS; MiniImageNet, VGG-Flower, and CUB. Our model inversion approach successfully inverts the essential discriminative features.

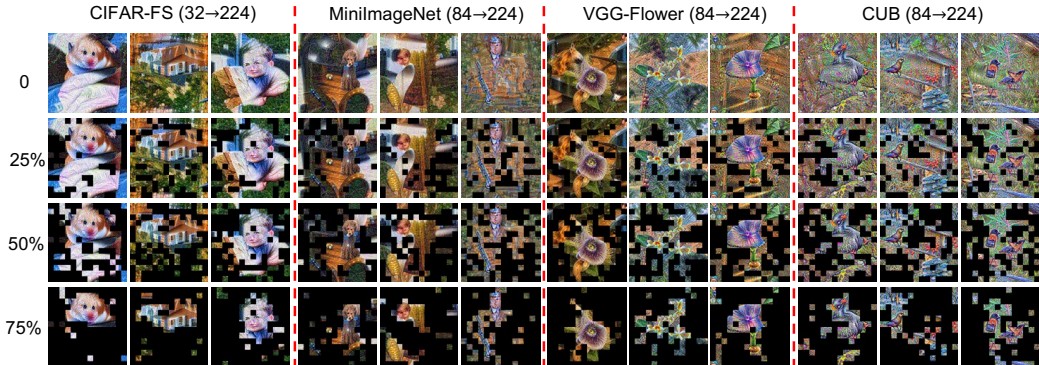

Figure 6: Visualization of masked synthetic images at varying sparsity levels. $(32 \rightarrow 224)$ denotes the original training images' resolution is $32 \times 32$ while we can reconstruct images with a higher resolution of $224 \times 224$.

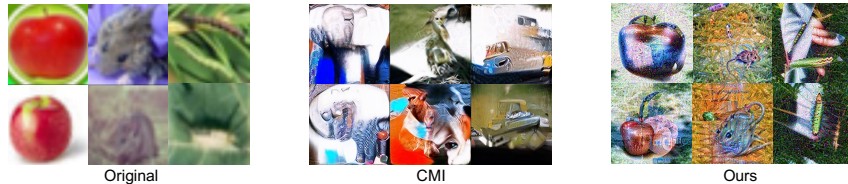

Figure 7: Comparison with SOTA model inversion approach. Our model inversion approach surpasses the current SOTA method CMI (Fang et al., 2021), delivering superior image quality with greater efficiency.

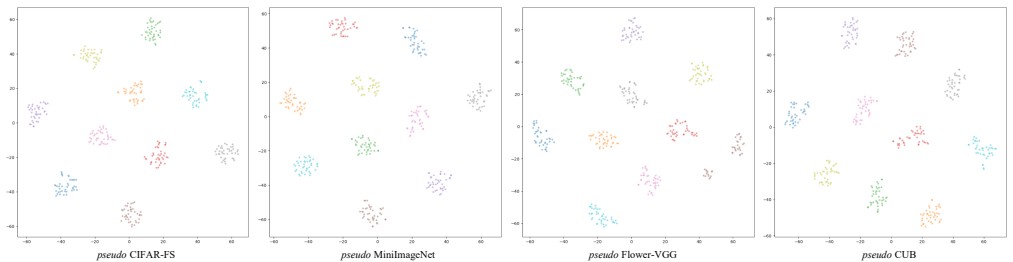

Figure 8: T-SNE visualization of synthetic images. Our model inversion approach successfully inverts the essential discriminative features, which is beneficial to the following meta-learning.

**Effect of cross-task interpolation.** Tab. 7 verifies the effectiveness of the cross-task interpolation under a constrained LoRA budget of 100 on CIFAR-FS. This technique can diversify the task distribution by generating multiple interpolated tasks, which enables the meta-training to cover a broader range of tasks, thereby bolstering the generalization capabilities for unseen tasks.

Table 7: Effect of cross-task interpolation.

| Ablation | 5-way 1-shot | 5-way 5-shot |
|---|---|---|
| **w/o** cross-task interpolation | 87.97 | 96.81 |
| **w/** cross-task interpolation | 89.69 | 97.05 |

**Effect of the naturalness prior.** Fig. 9 shows the efficacy of the regularization term $\mathcal{R}_{BN}$ in Eq. (1) to enhance the realism of images by enriching natural color and smoothing noise. We set the coefficient $\alpha_{BN}$ as 0.01.

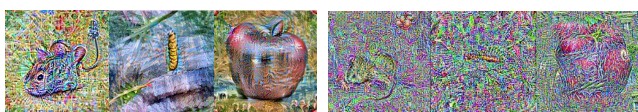

Figure 9: Visualization of synthetic images with (left) and without (right) the naturalness prior $\mathcal{R}_{BN}$.

**Meta-learn what?** Our framework meta-trains an extra lightweight LoRA while keeping the original VFM frozen. Based on the results shown in Tab. 8, we summarize some findings: (i) Meta-training the entire VFM is inferior to only meta-training the extra LoRA. Meta-training the entire VFM might distort the original feature space (Kumar et al., 2022), leading to bias to meta-training tasks and heavy costs of computation and storage. Meta-training the extra LoRAs can preserve the knowledge of foundation models learned from large-scale pretraining while injecting task-specific knowledge into extra LoRAs. (ii) Only meta-training the last 6 LoRA layers can outperform meta-training all LoRA layers. The improvements are more obvious in 5-way 1-shot learning, suggesting that reducing learnable parameters possibly avoids overfitting with limited training data. Only meta-training the first 6 LoRA layers is less effective. This is because only updating the shallow layers is insufficient to develop effective representations compared with updating the deep layers.

Table 8: Meta-learn what?

| Learnable Parts | 5-way 1-shot | 5-way 5-shot |
|---|---|---|
| Entire VFM | 88.40 | 95.73 |
| LoRA (all 12 layers) | 89.69 | **97.05** |
| LoRA (the first 6 layers) | 85.45 | 95.13 |
| LoRA (the last 6 layers) | **90.40** | 96.10 |

**Recycle LoRAs with different ranks.** Tab. 9 verifies the architecture-agnostic feature of our LoRA Recycle approach. Our approach can reuse pre-tuned LoRAs with different ranks (e.g., 50% LoRAs with the rank of 4 and 50% LoRAs with the rank of 8). This is a distinctive advantage absent in existing baselines, thereby extending its practical applicability across various real-world scenarios.

Table 9: Architecture-agnostic property of our framework. We conduct experiments on CIFAR-FS and set the rank of meta-LoRA as 4. We reuse pre-tuned LoRAs with different ranks (e.g., 50% LoRAs with the rank of 4 and 50% LoRAs with the rank of 8).

| Rank of pre-tuned LoRAs | 5-way 1-shot | 5-way 5-shot |
|---|---|---|
| 100%: **4** | 89.69 | 97.05 |
| 50%: **4** + 50%: **8** | 90.67 | 97.12 |

**Cross validation.** Tab. 10 shows our consistent superiority compared with other baselines by exchanging meta-training and meta-testing domains.

Table 10: Cross validation by exchanging meta-training and meta-testing domains. [meta-training domains]→[meta-testing domain]. $\mathcal{D}_1$: MiniImageNet, $\mathcal{D}_2$: CUB, $\mathcal{D}_3$: CropDiseases. 51: 5-way 1-shot. 55: 5-way 5-shot.

| Method | $[D_2, D_3] \rightarrow [D_1]$ | | $[D_1, D_3] \rightarrow [D_2]$ | | $[D_1, D_2] \rightarrow [D_3]$ | |
|---|---|---|---|---|---|---|
| | 51 | 55 | 51 | 55 | 51 | 55 |
| LoRAHub + NN | 81.02 | 93.18 | 85.27 | 95.23 | 76.21 | 92.31 |
| LoRA Recycle$_{75}$ (ours) | **86.12** | **95.03** | **90.02** | **97.12** | **80.19** | **94.02** |

**Experiments on more challenging dataset, Meta-Dataset** (Triantafillou et al., 2020). We evaluate our proposed LoRA Recycle framework on the Meta-Dataset, a benchmark specifically designed to test few-shot learning models across a variety of challenging domains. This dataset provides a rigorous evaluation setting. The results, summarized in Tab. 11, demonstrate the effectiveness of LoRA Recycle compared to other baselines. Notably, LoRA Recycle achieves superior performance in both the 5-way 1-shot and 5-way 5-shot settings, while also offering the advantage of being fine-tuning-free.

Table 11: Experiments on Meta-Dataset.

| Method | Fine-Tuning-Free | 5-way 1-shot | 5-way 5-shot |
|---|---|---|---|
| MOLE | ✗ | 61.87 | 76.31 |
| LoRAHub | ✗ | 63.14 | 77.24 |
| LoRA Recycle (ours) | ✓ | 68.48 | 80.12 |

**Experiments on tasks beyond few-shot learning.** In this section, we extend our evaluation to zero-shot classification tasks, demonstrating the versatility of LoRA Recycle beyond few-shot learning. To enable zero-shot classification, we recycle pre-tuned LoRAs from CLIP by replacing the classification loss used in Eq. (1) and Eq. (4) with the contrastive loss employed by CLIP. Tab. 12 presents the results on the Meta-Dataset for zero-shot classification. As shown, LoRA Recycle significantly outperforms other baseline methods, including MOLE and LoRAHub, both of which require fine-tuning. LoRA Recycle, being fine-tuning-free, achieves a higher accuracy, illustrating its effectiveness in adapting to zero-shot classification tasks.

Table 12: Experiments on zero-shot classification on Meta-Dataset.

| Method | Fine-Tuning-Free | 5-way 0-shot |
|---|---|---|
| MOLE | ✗ | 59.36 |
| LoRAHub | ✗ | 60.25 |
| LoRA Recycle (ours) | ✓ | 64.52 |

**Experiments on more types of Vision Transformers.** In this section, we evaluate the performance of our LoRA Recycle framework across multiple Vision Transformers on the CIFAR-FS dataset, further demonstrating its generalizability. We experiment with three popular Vision Transformer architectures: ViT-B (CLIP), DeiT-B (Touvron et al., 2021), and LV-ViT-M (Jiang et al., 2021). Each model is compared using LoRAHub as a baseline. Tab. 13 presents the results of these experiments. The performance is evaluated in both 5-way 1-shot and 5-way 5-shot scenarios. As shown, LoRA Recycle consistently outperforms the LoRAHub baseline, while also offering the advantage of being fine-tuning-free.

Table 13: Experiments on more types of Vision Transformers on CIFAR-FS.

| Model | Method | Fine-Tuning-Free | 5-way 1-shot | 5-way 5-shot |
|---|---|---|---|---|
| ViT-B (CLIP) | LoRAHub | ✗ | 81.02 | 96.24 |
| | LoRA Recycle (ours) | ✓ | 91.03 | 97.05 |
| DeiT-B (Touvron et al., 2021) | LoRAHub | ✗ | 79.52 | 93.32 |
| | LoRA Recycle (ours) | ✓ | 88.31 | 94.72 |
| LV-ViT-M (Jiang et al., 2021) | LoRAHub | ✗ | 80.42 | 94.23 |
| | LoRA Recycle (ours) | ✓ | 89.52 | 95.35 |

## B  PRELIMINARY OF VISION TRANSFORMERS (VITS)

**Preliminary of ViTs.** Here, we discuss the operational mechanism behind ViTs. ViTs initially divide the input image $X^{\mathrm{I}}$ belonging to the space $\mathbb{R}^{H \times W \times C}$ into $n + 1$ distinct, non-overlapping patches. These patches are then transformed into $n+1$ tokens, denoted as $X^{\mathrm{I}} = [x_{\texttt{[CLS]}}, x_1, ..., x_n]$ where $x_i \in \mathbb{R}^D$. The class token, $x_{\texttt{[CLS]}}$, is prepended to these image tokens to facilitate the classification task. To integrate positional relationships, learnable position encodings are added to all tokens. These tokens are then processed through multiple ViT layers, which are composed of multi-head self-attention (MHSA) modules and feed-forward networks (FFN). Within each MHSA, the token set $X^{\mathrm{I}}$ undergoes the transformation into three distinct matrices: the query $Q$, key $K$, and value $V$ matrices. The formulation of the attention mechanism is given by

$$\text{Attention}(Q, K, V) = \text{Softmax}\left(\frac{QK^T}{\sqrt{d}}\right) V, \tag{7}$$

where $d$ represents the dimension of the query vectors within $Q$. We define $A$ as the square matrix representing the attention weights across all token pairs, calculated as $A = \text{Softmax}\left(\frac{QK^T}{\sqrt{d}}\right)$, with dimensions $\mathbb{R}^{(n+1) \times (n+1)}$. Specifically, $a_i$, which is the $i^{\text{th}}$ row of $A$, signifies the attention weights of token $x_i$ with respect to all tokens. Particularly, $a_{\texttt{[CLS]}}$ refers to $a_0$. Based on Eq. (7), the $i^{\text{th}}$ output token can be viewed as a linear combination of all tokens' value vectors $[v_{\texttt{[CLS]}}, v_1, ..., v_L]$, weighted by $a_i$. These output tokens are subsequently forwarded to the FFN, which consists of two linear layers and an activation function. At the final ViT layer, the class token $x_{\texttt{[CLS]}}$, summarizing the global image representation, is utilized as the classifier's input to predict the image's classification probability distribution.

**Computational complexity of ViTs.** Given an image split into $N$ patches, each with an embedding dimension of $D$, the computational complexities of self-attention (SA) and feed-forward network (FFN) in ViTs are :

$$O(\text{SA}) = 3ND^2 + 2N^2D, \;\; O(\text{FFN}) = 8ND^2. \tag{8}$$

Since the complexities of SA and FFN scale respectively quadratically and linearly with $N$, our proposed double-efficient mechanism (see Sec. 4.3) significantly reduces the computational complexity by reducing the number of tokens.

## C    HYPERPARAMETER SELECTION AND SENSITIVITY ANALYSIS

In this section, we detail the selection of hyperparameters and conduct a sensitivity analysis on key hyperparameters. Generally speaking, We base our hyperparameter values on reference works and perform grid searches within the relevant ranges to identify the optimal configurations.

For the learning rate in LoRA Inversion, we refer to the settings from prior work (Yin et al., 2020), and perform a grid search over the range $[0.1, 0.25, 0.5]$. Similarly, for the learning rate in the meta-learning stage, we adopt values from the literature (Snell et al., 2017) and conduct a grid search over the range $[0.001, 0.01, 0.1]$. These ranges allow us to identify the optimal configurations.

We further conduct sensitivity analysis of the hyperparameter $\alpha_{\mathcal{R}}$ in Eq. (1), as it controls the balance during the inversion process. To analyze this, we conducted experiments on the CIFAR-FS dataset in both 5-way 1-shot and 5-way 5-shot settings. Tab. 14 shows the results, where we varied the value of $\alpha_{\mathcal{R}}$ to observe its effect on accuracy. Our sensitivity analysis reveals that our framework is not very sensitive to changes in $\alpha_{\mathcal{R}}$, although there are some variations among different $\alpha_{\mathcal{R}}$ values. This stability simplifies the hyperparameter tuning process, making our framework easier to apply in real-world applications.

Table 14: Sensitivity analysis of $\alpha_{\mathcal{R}}$ in Eq. (1) on CIFAR-FS.

| Hyperparameter | 5-way 1-shot | 5-way 5-shot |
|:--------------:|:------------:|:------------:|
| 0.1            | 89.35        | 96.39        |
| 0.01           | 89.70        | 96.69        |
| 0.001          | 88.83        | 95.76        |

## D    IMPLEMENTATION DETAILS OF BASELINES

Here, we provide detailed implementation details for the baselines used in our paper..

- **Fine-tuning baselines.** "Full Fine-Tuning" updates the entire model on the target task via gradient descent. "Linear probe" only updates the classification head. "LoRA + Linear (Hu et al., 2021)" updates the layer-wise rank decomposition matrices and the classification head. For fine-tuning, we select the best results from learning rates $[0.1, 0.01, 0.001]$. For LoRA, we set the rank to 4.

- **Multi-LoRAs composition baselines.** "LoRAs Avg" refers to averaging all given pre-tuned LoRAs into a single LoRA, which can be further fine-tuned with the classification head ("LoRAs Avg + Linear") or directly make inference via Nearest Neighbour ("LoRAs Avg + NN") without fine-tuning. "LoRAHub (Huang et al., 2023a)" takes a further step which obtains a single LoRA by a weighted sum of given pre-tuned LoRAs, where the weight values are fine-tuned on the target task. "MOLE (Chen et al., 2024)" fine-tunes a learnable gating function to composing the outputs of different LoRAs. For LoRAHub, we use a gradient-free approach to fine-tune the coefficients of pre-tuned LoRAs, following the setup in the original paper. For MOLE, we use gradient descent to fine-tune the learnable gating function. We select the best fine-tuning results from learning rates $[0.1, 0.01, 0.001]$.

- **Few-shot adaptation.** The current state-of-the-art baseline, $P > M > F$ (Hu et al., 2022), performs few-shot adaptation by stacking three stages: pre-training, meta-training, and fine-tuning. We follow the original paper's setup and apply data augmentation to the support set of the target tasks. We select the best fine-tuning results from learning rates $[0.1, 0.01, 0.001]$.

- **Fine-tuning-free baselines.** "Nearest Neighbour (NN)" makes predictions based on the label of the closest class center. "CAML (Fifty et al., 2023)" trains a sequence model to simulate the in-context learning of LLMs. Since we do not have real data to train the sequence model, we use synthetic data generated from pre-tuned LoRAs to train the sequence model. All other settings are consistent with the original paper.

## E    MORE DISCUSSIONS

**Discussions on the inconsistent performance gains across various datasets.** When we use Lo-RAs from the dataset the same as the testing dataset (in-domain setting), those LoRAs can provide domain-specific priors. This is particularly useful when the foundation model's pre-training dataset varies from the testing dataset. The main paper's Tab. 2 confirms this, showing a higher performance gain on CIFAR-FS (+10.01%) than other datasets (average +4.98%). The larger disparity between CIFAR-FS and the pre-training dataset is supported by the baseline NN in the main paper's Tab. 2, showing that directly transferring the foundation model to CIFAR-FS results in a lower accuracy (78.06%) compared to other testing datasets (average 85.31%). When we use LoRAs from datasets different from the testing dataset (cross-domain setting), performance gains across datasets are relatively stable, since these LoRAs offer limited useful domain-specific priors for all testing datasets.

**Paradigms for Adaptable Foundation Models** Several paradigms have been proposed to make large foundation models more adaptable. These paradigms involve combinations among Pre-training (P), Meta-learning (M), Fine-tuning (F) or PEFT, and In-context learning (I). Here, we provide a discussion over three paradigms, including P>F or P>PEFT, P>M>F and P>M>I. > indicates the sequence. Traditional P>F and P>PEFT (Fu et al., 2023b; Lee et al., 2023; Sun et al., 2019) often fail to adapt foundation models to data-limited and real-time applications due to their need for sufficient data and explicit fine-tuning.

An emerging strategy, P>M>F, introduces a meta-learning phase before fine-tuning, preparing the pre-trained model for subsequent fine-tuning. This paradigm has shown promising results in vision (Hu et al., 2022; Cai & Shen, 2020), language (Gheini et al., 2023; Bansal et al., 2022; Hou et al., 2022) and vision-language (Yeh et al., 2023; Najdenkoska et al., 2023; Huang et al., 2023b) domains.

More recently, the P>M>I paradigm has been proposed in language domains, aiming to acquire more advanced in-context learning ability of LLMs. For example, LLMs are equipped with the instruction-following ability by meta-training on a broad range of tasks accompanied by instructions (Iyer et al., 2022; Chung et al., 2022). MetaICL (Min et al., 2022) and ICT (Chen et al., 2022) explicitly meta-train LLMs to learn to learn in context. However, paradigms for fine-tuning-free adaptation in VFMs are less explored, hindered by their inherent in-context learning limitations compared to LLMs.

## F    RETHINKING EXISTING DATA-FREE META-LEARNING METHODS

**Limited scalability to large-scale models.** Current data-free meta-learning (DFML) methodologies, as discussed in (Wang et al., 2022; Hu et al., 2023a;b), predominantly focus on leveraging small-scale pre-trained models and meta-learners, such as four-layer CNNs or ResNet12. A critical limitation of these approaches is their inability to scale up to larger models, particularly those based on transformer architectures. This scalability issue substantially hinders their practical application in complex, real-world scenarios. For instance, (Wang et al., 2022) employs a hyper-network with all pre-trained models as inputs and outputs a single fused model. The efficiency of this method declines significantly when outputting all parameters of larger models, given the hyper-network's extensive input and output dimensions. Similarly, inversion-based DFML methods, such as those in (Hu et al., 2023a;b), rely on meta-training a meta-learner with data inverted from pre-trained models. The model inversion process becomes inefficient for large-scale models. The following meta-training process often necessitates the computation of Hessian matrices for second-order derivatives (Nichol et al., 2018), which becomes exceedingly resource-intensive for large-scale models.

**Inefficiency issues.** Beyond scalability challenges, inversion-based methods like (Hu et al., 2023a;b) are plagued by the inefficiency of the model inversion processes. These methods typically involve

iterative forward and backward optimizations, leading to significant computational and storage costs when applied to large-scale models.

**Addressing these issues: our contributions.** Our Double-Efficient Data-Free Meta-Learning framework presents a novel and efficient solution that is scalable for transformer-based foundation models. To overcome existing limitations, (i) we propose a data-free meta-learning framework, which is specifically designed for large-scale VFM. We only meta-train meta-LoRA, constituting only 0.14M parameters (merely 0.1% relative to the VFM). (ii) We propose a meta-learning objective, as outlined in Eq. (3), that avoids the resource-intensive computation of Hessian matrices. This is achieved as our fine-tuning-free adaptation in the inner loop does not require gradient computations. (iii) We propose a double-efficient mechanism that significantly speeds up the meta-training processes while maintaining comparable or enhanced performance. Our approach not only addresses the limited scalability and inefficiency issues of existing DFML methods, but also inspires more interactions between meta-learning and foundation models.

