# OpenReview forum: "LoRA Recycle: Towards Fine-Tuning-Free Visual Foundation Model via Double-Efficient Data-Free Meta-Learning"
_ICLR.cc/2025/Conference — ICLR 2025 Conference Withdrawn Submission_

### Official Review · Reviewer_rHzM · 2024-10-22

**Soundness:** 2
**Presentation:** 3
**Contribution:** 2
**Rating:** 3
**Confidence:** 2

**Summary:**

Vision foundation models often require fine-tuning with large data to perform well on a few shot tasks. Develop a real-time few-shot system with minimal data in real-time. Existing LoRA techniques require fine-tuning, which makes them unsuitable for real-time response, and a large training dataset causes instability at a small scale.
The work proposes: "data free" recycling existing LoRA modules to achieve impressive few-shot performance.

**Strengths:**

1.	Paper is written in easy to understand manner.

2.	The 5-way 1-shot accuracy improvement is impressive, thus proving the proposed methods utility.

3.	Visualization provided makes understanding synthetic dataset easy. Figure 2 and Figure 3 is really well made, makes understanding paper easy.

4.	Masking images as a means of computation efficiency is an interesting idea. As well as using the self-attention weights for pruning tokens is interesting too.

**Weaknesses:**

1.	**Mentioning terms without definition**, [LINE 023] “meta-LoRA” [Line 024] “LoRA Inversion”. Maybe make them italics to show emphasis as a standard procedure.

2. When comparing with existing methods, **missing work** include
a.	*fine-tuning  “Visual prompt tuning”* (Jia, Menglin, et al. "Visual prompt tuning." European Conference on Computer Vision. Cham: Springer Nature Switzerland, 2022).
b.	*LoRA “The Balanced-Pairwise-Affinities Feature Transform”* (Shalam, Daniel, and Simon Korman. "The Balanced-Pairwise-Affinities Feature Transform." Forty-first International Conference on Machine Learning (2024).)
c. *Efficinet technique like Test-time prompt tuning*  (Shu, Manli, et al. "Test-time prompt tuning for zero-shot generalization in vision-language models." Advances in Neural Information Processing Systems 35 (2022): 14274-14289)


3. **Results looks unconvincing**. Take the baselines, “LoRAHub” “MOLE” and “LoRAs Avg + Linear” their “5-way 5-shot” performance is similar to “LoRA Recycle” (inferior by 1%). These baselines are far more computationally efficient (no synthetic dataset generation and no distillation), yet give comparable performance. While LoRA Recycle performs well in the “5-way 1-shot” setting, the method doesn’t seem to highlight any special technique/method that helps in this particular result. It appears to be an unintentional benefit of the proposed method. This is more prominent in cross-domain results (Table 3).

4. **Key Motivations are missing**:
(a)	Why are the authors using synthetic data (“Data-free” & “avoids additional data collection”)? What’s the motivation behind it? What happens if the model uses any standard dataset like “MiniImageNet” on which these LoRA(s) are already pre-trained on (in-domain)
(b) **Line [045] “leads to significant time overheads and increased memory usage.”** Generating synthetic dataset has a significant computation / time overhead as well. How is using synthetic dataset a better alternative than using a large scale dataset like Laion-2b as unsed in CAML? If It were to assume, synthetic images are noisy making them sparse (removing tokens) would reduce the noise and improve performance as observed in the ablation.
(c) What's the motivation behind using LoRAs? Other methods like prompt tunning, test time augmentation, etc. are not beneficial. The technique doesn’t compare these methodologies and determining the utility of LoRAs in isolation is difficult and not well-motivated.
(d) [Line 084] “parameter-lightweight” and “computation-efficient” [Line 085]? This approach is not lightweight, as it needs to account for the “trainable pixels” that need to be trained during LoRA inversion for generating a synthetic dataset thereby giving it a data-free status.
(e)	[Line 086] “architecture agnostic, enabling to recycle LoRAs with heterogeneous architectures like different ranks, as a distinct advantage over existing methods.” Is it? The synthetic dataset is generated based on a gradient from VFMs. The proposed solution is based on the choice VFM. if this is still considered as architecture agnostic, most existing fine-tuning techniques like adapters and promotes are architectural agnostic.

5. **Key solutions are not solving the motivation**: The solution is to propose a real-time fine-tuning module for few-shot learning.
(a) *Generating synthetic* data solves the data-free problem? Ablation needs to show what happens if the standard dataset like Laion-2b is used to motivate the use of synthetic dataset (and answer the data-free problem)
(b) *Training in retrieval-based technique* (Line[228]  Synthetic few-shot task construction): Are authors claiming retrieval-based techniques help in Line[036] "few-shot tasks without the necessity for fine-tuning,"

**Questions:**

Please address all the weakness mentioned above

---

> ### Author Response · Authors · 2024-11-15
> **Author Rebuttal (1/1)**
>
> > Q1: Discussion on the missing work.
>
> A1: Thank you for the suggestion. We would like to clarify the rationale and comprehensiveness of our baseline choices. Our work primarily focuses on achieving tuning-free few-shot adaptation through LoRA reuse, rather than outperforming all fine-tuning methods such as (a) visual prompt tuning, (b) The Balanced-Pairwise-Affinities Feature Transform, and (c) test-time prompt tuning. Instead, we aim to highlight the unique advantages of our tuning-free approach, as presented in Tab. 1 of the manuscript. Tuning-free methods like ours are better suited for real-time applications and exhibit greater stability with extremely limited tuning data (e.g., in 1-shot scenarios).
>
> ------
>
> > Q2: (1) Results on 5-way 5-shot learning. (2) Computational efficiency considerations.
>
> A2: (1) We acknowledge that for certain configurations, such as the 5-way 5-shot setting, the performance gap between *LoRA Recycle* and some baselines is narrower. However, we would like to emphasize that *LoRA Recycle* is specifically designed as a tuning-free approach, which provides unique advantages in real-time applications. Additionally, our method demonstrates a notable advantage in 1-shot scenarios, outperforming other baselines by an average of 6.27%.
>
> (2) We understand your concern regarding computational efficiency, as LoRA Recycle involves synthetic data generation. However, it is important to note that data synthesis and meta-training are one-time processes. Once our meta-LoRA is meta-trained, it can be deployed across diverse downstream tasks without requiring further fine-tuning. In contrast, fine-tuning-based methods need to be tuned separately for each new task.
>
> ------
>
> > Q3: Why are the authors using synthetic data?
>
> A3: Thank you for your question. Synthetic data is used because users typically share only the trained LoRA weights, without providing access to the original training data, often due to privacy concerns. By using synthetic data, we can effectively substitute for the unavailable original data, enabling our approach to work within these privacy constraints.
>
> ------
>
> > Q4: Generating synthetic dataset has a significant computation / time overhead as well.
>
> A4: It’s important to note that data synthesis is a one-time process. Once meta-training is completed using the synthetic data, the resulting meta-LoRA can be directly applied to diverse downstream tasks without further fine-tuning. In contrast, traditional fine-tuning methods require separate tuning for each new task, making them unsuitable for real-time applications and often unstable when tuning data is limited.
>
> ------
>
> > Q5: What's the motivation behind using LoRAs?
>
> A5: Our motivation for using LoRA lies in its lightweight nature and proven effectiveness, making it a well-established and widely recognized parameter-efficient fine-tuning method. While our experiments focus on LoRA reuse, the framework could also be adapted to work with other modular fine-tuning methods highlighting the versatility of our proposed solution.
>
> ------
>
> > Q6: clarification of "architecture agnostic".
>
> A6: By "architecture agnostic," we mean that our LoRA reuse method can accommodate LoRAs with different ranks. This is a unique advantage compared to previous LoRA reuse methods, which are often limited to parameter arithmetic and therefore require strictly identical LoRA structures.

---

### Official Review · Reviewer_t9m3 · 2024-11-01

**Soundness:** 2
**Presentation:** 3
**Contribution:** 2
**Rating:** 5
**Confidence:** 4

**Summary:**

This paper addresses the challenge of extra costs and limited resources when adapting large-scale visual models to different domains, focusing on classification. The proposed method employs meta-learning to develop a meta-LoRA capable of performing classification in a single forward pass. The authors validate their approach through experiments on several datasets in a few-shot setting.

**Strengths:**

1. The idea of distilling knowledge from various pre-finetuned LoRAs to achieve generalized understanding without requiring access to the original datasets is intriguing.
2. The authors provide clear explanations of their methods, making the paper easy to follow.

**Weaknesses:**

1. The generalizability of the proposed method is questionable, as the experiments were conducted on only eight small datasets. While out-of-domain experiments were performed, the results on the ISIC and CHESTX-RAY datasets were unsatisfactory, possibly due to limited category diversity.
2. Although the motivation for the proposed method is compelling, the authors did not utilize a wide range of pre-finetuned LoRAs from the community. Instead, they constructed datasets from existing ones, which is not entirely convincing.
3. More comparative methods should be included, such as CooP, CoCOOP, and PromptSRC, to provide a more comprehensive evaluation.

**Questions:**

See weakness

---

> ### Author Response · Authors · 2024-11-15
> **Author Rebuttal (1/1)**
>
> > Q1: The results on ISIC and CHESTX-RAY datasets.
>
> A1: The lower performance on the ISIC and CHESTX-RAY datasets is due to the significant domain gap between these medical imaging datasets and the natural image domain of the training set. As seen from the baseline results, even state-of-the-art methods struggle with these datasets. Despite this, LoRA Recycle achieves notable improvements, surpassing the best fine-tuning-based baselines by up to 4.31% and 2.41% for 1-shot and 5-shot learning, respectively. It also outperforms the leading tuning-free baselines by up to 4.11% and 7.02% for 1-shot and 5-shot learning. While LoRA Recycle demonstrates substantial performance gains, further improvements are possible in handling large distribution shifts, highlighting a promising direction for future research.
>
> ------
>
> > Q2: Utilize a wide range of pre-finetuned LoRAs from the community.
>
> A2: Currently, in the vision domain, there is no established benchmark that consolidates a comprehensive set of pre-trained LoRAs. Therefore, we constructed our own set of pre-trained LoRAs from existing datasets. We plan to release our LoRA library as an open benchmark to facilitate future research and provide a standardized resource for the community.
>
> ------
>
> > Q3: More comparative methods should be included, such as CooP, CoCOOP, and PromptSRC
>
> A3: Thank you for your suggestion to include additional comparative methods such as CooP, CoCOOP, and PromptSRC. We acknowledge the variety of fine-tuning approaches available; however, we would like to emphasize a few key points:
>
> 1. Our goal is not necessarily to surpass the performance of fine-tuning methods, but rather to highlight the unique advantages of our tuning-free approach, as shown in Tab. 1 of the manuscript. Tuning-free methods like ours are better suited for real-time applications and demonstrate greater stability with extremely limited tuning data (e.g., 1-shot scenarios).
> 2. Additionally, CooP, CoCOOP, and PromptSRC are specifically designed for use with CLIP and rely on category names as text prompts. In contrast, our focus is on few-shot classification without the need for class-specific textual prompts.

---

### Official Review · Reviewer_94Fr · 2024-11-03

**Soundness:** 3
**Presentation:** 3
**Contribution:** 2
**Rating:** 5
**Confidence:** 4

**Summary:**

This paper addresses the challenge of reusing existing LoRAs for adapting a new VFMs to few-shot tasks without the need of original data or tuning. To achieve this, the authors propose data-free meta-learning framework. By distilling a meta-LoRA using synthetic data from LoRA Inversion, the framework enables VFMs to perform few-shot tasks in a single pass, similar to in-context learning in LLMs. Additionally, a double-efficient mechanism accelerates meta-training by focusing on foreground patches, enhancing both speed and performance. Extensive evaluations across several datasets demonstrate the framework’s effectiveness in both in-domain and cross-domain scenarios.

**Strengths:**

1. This work introduces an interesting task to explore the potential of reusing diverse pre-tuned LoRAs, expanding the utility of these modules beyond traditional task-specific applications.
2. The paper is well written and easy to follow.
3. The proposed method performs well on several datasets.

**Weaknesses:**

1. Related Work: The paper lacks a thorough discussion on data-free knowledge distillation.

2. Limited Novelty: While the paper attempts to tackle a novel and interesting problem, the techniques employed to address it appear somewhat basic and lack innovation. The authors suggest inverting LoRA to obtain synthetic data, a standard approach commonly used in data-free KD literature. Additionally, the model training relies on basic meta-learning methods combined with ProtoNet, a technique widely applied in few-shot learning research. There does not appear to be any unique techniques specifically proposed for LoRA recycling. Furthermore, it seems plausible that this approach could be generalized to recycle various models, not just LoRA, without significant modification to the methodology. This raises questions about the uniqueness and specificity of the proposed solution. The authors could refer to the paper for a similar method: https://arxiv.org/pdf/2110.04545.

3. Limited Evaluation: The evaluation uses relatively simple, toy datasets, which may not fully showcase the robustness or generalizability of the proposed approach. To strengthen the evaluation, I recommend including more challenging datasets, such as WILDS or DomainNet, which could better test the model's performance in diverse, real-world scenarios.

4. Ablation Study: The necessity of meta-learning is unclear. An ablation study focusing on the role of meta-learning would provide valuable insights into its contribution and justify its inclusion in the model.

**Questions:**

1. Patch Masking vs. Token Reduction: Why is masking patches chosen over reducing the number of tokens in synthetic data generation? An explanation of the design choice here could clarify its benefits and relevance to the overall model.

2. Typo: Line 090L has a typo: "re-quiring" should be corrected to "requiring."

---

> ### Author Response · Authors · 2024-11-15
> **Author Rebuttal (1/2)**
>
> > Q1: Related Work: The paper lacks a thorough discussion on data-free knowledge distillation.
>
> A1: We acknowledge the relevance of our approach to data-free knowledge distillation (DFKD), as both methods utilize model inversion techniques to generate data. We will include a discussion of related work in the revised version.
>
> Data-Free Knowledge Distillation (DFKD) transfers knowledge from a large, pre-trained teacher model to a smaller, lightweight student model without using the original training data. This approach is particularly valuable in contexts where data access is restricted due to privacy or ethical concerns. Influenced by model inversion techniques, DFKD methods such as DeepInversion [1] and CMI [2] generate synthetic images by leveraging teacher model statistics and classification objectives to enhance knowledge transfer. Additionally, ABD [3] explores the security risks in DFKD, particularly focusing on backdoors. DFKD methods have also found applications in federated learning [4], model quantization [5], and other areas.
>
> In contrast to DFKD, which uses inverted data to distill knowledge from a single teacher network, our work leverages inverted data for meta-learning across multiple teacher networks. Furthermore, we propose a double-efficient mechanism that accelerates both the inversion and meta-training processes.
>
> [1] Dreaming to Distill: Data-free Knowledge Transfer via DeepInversion, CVPR 2020
>
> [2] Contrastive Model Inversion for Data-Free Knowledge Distillation, IJCAI 2021
>
> [3] Revisiting Data-Free Knowledge Distillation with Poisoned Teachers, ICML 2023
>
> [4] Data-Free Knowledge Distillation for Heterogeneous Federated Learning, ICML 2021
>
> [5] Patch Similarity Aware Data-Free Quantization for Vision Transformers, ECCV 2022
>
> ------
>
> > Q2: (1) Novelty and Contribution of Our Approach. (2) Potential Generalizability Beyond LoRA. (3) Difference from the Suggested Paper.
>
> A2: Thank you for your thoughtful feedback. We would like to address each of your points below:
>
> 1. **Novelty and Contribution of Our Approach**:
>    To our knowledge, this is the first work to explore reusing pre-tuned LoRAs for achieving tuning-free few-shot adaptation in VFMs, offering new insights into leveraging the accessibility and diversity of pre-tuned LoRAs beyond traditional task-specific reuse. The primary novelty of our contribution lies not in developing new meta-learning or model inversion techniques, but in creatively applying their principles to design a novel LoRA reuse framework that addresses a new and interesting problem.
>
> 2. **Potential Generalizability Beyond LoRA**:
>    While our experiments focus on LoRA reuse, the framework could be adapted to work with other modular fine-tuning methods as well. This generalizability is a strength of our approach, as it suggests broader applicability to various model adaptation scenarios, further underscoring the versatility of our proposed solution.
>
> 3. **Difference from the Suggested Paper**:
>    The suggested paper on data-free domain generalization differs from our work in both setting and technique:
>
>    - **Setting Differences**: Data-free domain generalization aims to learn domain-agnostic knowledge across multiple domains and transfer it to an unknown domain, requiring the training and testing domains to share the same label space, without necessitating few-shot scenarios. Our setting is more challenging: we learn prior knowledge from multiple LoRAs (tasks) and generalize to unseen tasks where the domains of training and testing tasks can differ, the label spaces between training and testing are strictly disjoint, and the testing tasks are in a few-shot context.
>
>      |   | Domain difference | Label difference | Few shot | Data Free |
>      | - | - | - | - | - |
>      | Data-Free Domain generalization | √   | X                | X        | √         |
>      | Our setting | √   | √     | √      | √         |
>
>    - **Technical Differences**: The method in the suggested paper relies on cross-domain model inversion to generate synthetic data with domain-agnostic features, which requires label space strictly same across domains, making it incompatible with our setting. In contrast, our approach leverages meta-learning principles, which do not impose this constraint, allowing us to generalize across tasks with differing label spaces.
>
> ------
>
> > Q3: Limited Evaluation on more challenging dataset.
>
> A3: We would like to point out that in Appendix A (Lines 906–917), we provide experimental results on the Meta-Dataset. Meta-Dataset is a more challenging benchmark, consisting of a collection of ten datasets across diverse domains, specifically designed to evaluate the generalization ability of few-shot learning models across various challenging scenarios. These results offer additional insights into the robustness of our approach on a benchmark created for testing model adaptability in complex and cross-domain contexts.
>
> Thank you again for your helpful feedback.
>
> ------

---

> ### Author Response · Authors · 2024-11-15
> **Author Rebuttal (2/2)**
>
> ------
>
> > Q4: Ablation Study: The necessity of meta-learning is unclear. An ablation study focusing on the role of meta-learning would provide valuable insights into its contribution and justify its inclusion in the model.
>
> A4: Meta-learning offers the advantage of producing a more generalizable model compared to conventional learning methods, such as supervised learning. To demonstrate this, we conduct an ablation study comparing meta-learning with joint supervised training on generated data. In the joint supervised training setup, we combine all task data for supervised learning. Evaluation is conducted  on unseen few-shot tasks. As shown in the table, we found that joint supervised training failed to generalize effectively to unseen few-shot tasks, whereas meta-learning, with its bi-level optimization, naturally learns a model with strong generalization ability.
>
> | Ablation                  | 5-way 1-shot on CIFAR-FS |
> | ------------------------- | ------------------------ |
> | Joint Supervised Learning | 78.22                    |
> | Meta-Learning             | 89.59                    |
>
> ------
>
> > Q5: Why is masking patches chosen over reducing the number of tokens in synthetic data generation?
>
> A5: In our approach, masking patches is equivalent to token pruning, as masking discards the masked tokens by excluding them from the forward and backward passes. This operates as a direct reduction in the number of active tokens.

---

### Official Review · Reviewer_Z1Vw · 2024-11-04

**Soundness:** 2
**Presentation:** 3
**Contribution:** 3
**Rating:** 5
**Confidence:** 4

**Summary:**

This paper proposed to reuse of pre-tuned LoRA techniques without the accessibility to the private data. This proposed method aims to improve the few-shot adaptability of VFMs without further fine-tuning and proposes a new data-free meta-learning framework. The experimental results on 8 datasets show the proposed method exceeds the existing literature.

**Strengths:**

1. Outstanding work on accelerating the meta-training process.
2. Efficient data synthesis with token pruning and meta-training with sparse tokens do a great job helping the generating and meta-learning process.

**Weaknesses:**

1. Based on the Lora market, this paper doesn’t get enough contribution to the pretrained Lora reusing method. The meta-learning is widely used in generalization problems, including zero-shot or few-shot learning tasks. The major contribution is not thus interesting to me.
2. Sparse tokens may break the potential correlation between foreground objects and background, this paper can’t simply think highly of this method without eliminating this potential adverse effect.
3. The token pruning in the data-efficient mechanism can also be found in other lightweight designs. Besides, I hope the authors highlight why this method is distinctive, especially when we only have generated data but not customized private data. In other words, what is the key relationship between them?
4. Several presentations are not clear or with several typos. e.g., Line 074, LoRs should be LoRAs.

**Questions:**

Please refer to the weakness section, my major concerns exist in the main contribution. The proposed techniques can be also widely found in other computer vision or LoRA architecture-designed papers. The authors should clearly claim why these proposed ideas contribute to this community.

The other major concern is about the relationship between this setting and cross-domain generalization. I wonder how the domain generalization method performs on this task. It seems these method could also focuses on the metra-learning techniques.

Besides, the usage of synthetic datasets could also show a clear upper bound. Thus the authors should discuss this and the relationship between using sufficient private datasets. Or in several extreme cases, what would happen, if there were several few-shot samples available? Is there a trade-off in these application scenarios?

---

> ### Author Response · Authors · 2024-11-15
> **Author Rebuttal (1/2)**
>
> > Q1: The contributions regarding LoRA reuse and meta-learning
>
> A1: We acknowledge that meta-learning has been widely applied to generalization problems. However, our contribution is distinct in its focus:
>
> - To the best of our knowledge, we are the first to propose a meta-learning-based LoRA reuse method.
> - Additionally, our study is the first to explore how LoRA reuse can enhance tuning-free few-shot adaptability of visual foundation models.
> - Our primary contribution lies not in creating a new meta-learning method, but in pioneering the use of meta-learning principles to design a LoRA reuse strategy and, for the first time, showing how LoRA reuse can enhance tuning-free few-shot adaptability in visual foundation models.
>
> ------
>
> > Q2: Sparse tokens may break the potential correlation between foreground objects and background.
>
> A2: Thank you for raising this point. While it's true that token pruning could disrupt correlations between foreground and background, our method specifically addresses this concern by leveraging the **unique characteristics of generated data**. When using inversion to generate data, the inversion process primarily crafts label-relevant features (typically the foreground regions) into the generated data while **background regions often remain noisy as initialization**. To further illustrate this, we have added an experiment below showing that, during the inversion process, the classification loss caused by the identified foreground steadily decreases, whereas the identified background loss remains nearly unchanged. As such, masking the background in generated data is unlikely to disrupt meaningful correlations. Instead, it helps to reduce noise, which aligns with our experimental results: pruning background tokens in the generated CIFAR-FS data led to a 1.34% improvement in performance.
>
> | Ablation             | Change of $\mathcal{L}_{\rm cls}$ during inversion |
> | - | - |
> | Inverted Backgrounds | 10.78 $\rightarrow$ 10.72                          |
> | Inverted Foregrounds | 10.78 $\rightarrow$ 0.12                           |
>
> ------
>
> > Q3: The uniqueness of token pruning in proposed framework. And the distinct advantages of token pruning when dealing with generated data, as opposed to customized private data.
>
> A3:
>
> - We acknowledge that token pruning is widely used in lightweight designs. However, our token pruning is distinctive in several ways:
>
>   (i) **New Application**: To the best of our knowledge, ours is the first work to use token pruning specifically to accelerate model inversion and meta-learning.
>
>   (ii) **Unique cross-stage pruning strategy tailored to our framework** :
>
>   **Inversion stage pruning:** During the inversion stage, token pruning is applied in the hidden layers by removing unimportant tokens based on self-attention weights, accelerating both forward and backward computations for data generation.
>
>   **Meta-training stage pruning:** **The pruning results from the inversion stage can guide the pruning for the following meta-training.** This can significantly speed up meta-training by early-layer pruning. Specifically, we construct a mask by setting values of 1 at the positions of remaining tokens and 0 elsewhere. We multiply the mask with the generated image to create a masked image. We then exclusively use the unmasked tokens for the following meta-training stage.
>
> - **Token pruning on generated data offers unique advantages beyond acceleration.** As noted in A2, the backgrounds in generated data tend to be noisy as initialization. Pruning these background regions helps reduce noise, contributing to meaningful performance improvements: pruning background tokens in the generated CIFAR-FS data led to a 1.34% improvement in performance.
>
> ------

---

> ### Author Response · Authors · 2024-11-15
> **Author Rebuttal (2/2)**
>
> > Q4: The relationship between this setting and domain generalization.
>
> Our setting is fundamentally different from domain generalization in several key aspects:
>
> Domain Generalization aims to learn across multiple domains to generalize to an unseen domain. It requires that both known and unseen domains share the same label space. For example, training domain 1 may include real images of cats and dogs, and training domain 2 may include animated images of cats and dogs. Then, the test domain would include paintings of cats and dogs.
>
> Our setting is more challenging, as both the labels and domains for training and testing tasks differ. Moreover, the labels in the test tasks are unseen during training. For example, in our setting, Task 1 might involve real images of cats and dogs, Task 2 might involve animated images of tigers and lions, and the test task could involve paintings of chairs and tables.
>
> Additionally, unlike domain generalization, our setting emphasizes a few-shot scenario in test tasks and does not require original data in training tasks.
>
> |                       | Domain difference | Label difference | Few shot | Data Free  |
> | --------------------- | ----------------- | ---------------- | -------- | ---------- |
> | Domain generalization | √                 | X                | X        | Not always |
> | Our setting           | √                 | √                | √        | √          |
>
> ------
>
> > Q5: (1) Discussion on the upper bound when using synthetic data. (2) Discussion on the case when few-shot examples are available. (3) Discussion on the trade-off between using synthetic and real original training data.
>
> Thank you for these thoughtful suggestions.
>
> - (1) We acknowledge that using synthetic data imposes an upper bound on performance, which is comparable to results achieved with each LoRA’s original training data. This is reasonable because the inversion process to generate synthetic data, is designed to approximate the original training data distribution as closely as possible.
> - (2) If few-shot examples from the original training data are available for each pre-tuned LoRA, these could guide the inversion process to produce synthetic data more closely aligned with the original data distribution, potentially enhancing performance.
> - (3) While real original training data could indeed yield better results, synthetic data offers a practical alternative that circumvents accessibility constraints related to the privacy protection, data labeling, and collection costs associated with original training data.

---

### Note · Authors · 2024-11-15

I have read and agree with the venue's withdrawal policy on behalf of myself and my co-authors.